# Mitochondrial ubiquinone–mediated longevity is marked by reduced cytoplasmic mRNA translation

Marte Molenaars[1], Georges E Janssens[1], Toon Santermans[2], Marco Lezzerini[1], Rob Jelier[2], Alyson W MacInnes[1], Riekelt H Houtkooper[1]

Mutations in the *clk-1* gene impair mitochondrial ubiquinone biosynthesis and extend the lifespan in *Caenorhabditis elegans*. We demonstrate here that this life extension is linked to the repression of cytoplasmic mRNA translation, independent of the alleged nuclear form of CLK-1. *Clk-1* mutations inhibit polyribosome formation similarly to *daf-2* mutations that dampen insulin signaling. Comparisons of total versus polysomal RNAs in *clk-1(qm30)* mutants reveal a reduction in the translational efficiencies of mRNAs coding for elements of the translation machinery and an increase in those coding for the oxidative phosphorylation and autophagy pathways. Knocking down the transcription initiation factor TATA-binding protein-associated factor 4, a protein that becomes sequestered in the cytoplasm during early embryogenesis to induce transcriptional silencing, ameliorates the *clk-1* inhibition of polyribosome formation. These results underscore a prominent role for the repression of cytoplasmic protein synthesis in eukaryotic lifespan extension and suggest that mutations impairing mitochondrial function are able to exploit this repression similarly to reductions of insulin signaling. Moreover, this report reveals an unexpected role for TATA-binding protein-associated factor 4 as a repressor of polyribosome formation when ubiquinone biosynthesis is compromised.

## Introduction

Slowing down mitochondrial metabolism is a well-known means of increasing the lifespan of multiple species (Dillin et al, 2002; Copeland et al, 2009; Houtkooper et al, 2013; Breitenbach et al, 2014). In *C. elegans*, one such model is the *clk-1(qm30)* mutant that harbors a deletion in the gene encoding a ubiquinone (UQ) biosynthesis enzyme (Felkai et al, 1999). UQ is a redox-active lipid that plays a central role in the electron transport chain and mitochondrial oxidative phosphorylation (OXPHOS) by carrying electrons from complexes I and II to complex III (Stefely & Pagliarini, 2017). In

addition to lifespan extension and decreased respiration, *clk-1 (qm30)* mutants are developmentally delayed and have reduced progeny (Felkai et al, 1999). Mammalian *Clk1,* known as Mclk-1 or Coq7, is also involved in the lifespan (Liu et al, 2005; Stepanyan et al, 2006; Takahashi et al, 2014). Transgenic mice, rescued from embryonic lethality via the transgenic expression of mouse clk-1, with reduced *Clk1* expression live longer and have smaller bodies than WT mice, demonstrating a conserved role for *clk-1* in longevity across species (Takahashi et al, 2014). However, it is not fully understood how the reduction of UQ biosynthesis extends the lifespan of these animals.

Until recently, CLK-1 protein was thought to reside exclusively in the mitochondria. However, a distinct nuclear form of CLK-1 has been reported to independently regulate lifespan by mitochondrial–nuclear retrograde signaling (Monaghan et al, 2015). Adding back a copy of the *clk-1* gene that lacks the mitochondrial targeting signal (MTS) to the *clk-1* deletion mutant led to a subtle but significant reduction in the lifespan extension phenotype observed in the full *clk-1* deletion mutant (Monaghan et al, 2015). However, the existence and function of a nuclear form of CLK-1 remains controversial. Other studies fail to detect any nuclear localization of the MCLK1/CLK-1 proteins or any biological activity of a *C. elegans* CLK-1 protein devoid of an MTS (Liu et al, 2017).

The importance of communication between stressed mitochondria and the rest of the cell is becoming increasingly appreciated (Quiros et al, 2016). One such reaction to stress is carried out by the mitochondrial unfolded protein response (UPR$^{mt}$) (Durieux et al, 2011; Baker et al, 2012; Houtkooper et al, 2013). GCN-2, an eIF2$\alpha$ kinase that modulates cytosolic protein synthesis, is phosphorylated in response to the activated UPR$^{mt}$ in *clk-1(qm30)* mutants in a manner required for lifespan extension (Baker et al, 2012). Another nonmitochondrial factor, TATA-binding protein-associated factor 4 (TAF-4), is a transcription factor that has been implicated in the extension of *clk-1 (qm30)* mutant lifespan (Khan et al, 2013). TAF-4 was identified in an RNAi screen for transcription factors required for the lifespan extension of *clk-1(qm30)* mutants (Khan et al, 2013). Beyond *clk-1*, TAF-4 was also required for the lifespan extension phenotype of two mutants of the mitochondrial electron transport chain (*isp-1* and *tpk-1*) (Khan et al, 2013). TAF-4 is a component of the transcription factor II D mRNA

[1]Laboratory Genetic Metabolic Diseases, Amsterdam University Medical Centers, University of Amsterdam, Amsterdam Gastroenterology and Metabolism, Amsterdam, The Netherlands  [2]Centre of Microbial and Plant Genetics University of Leuven, Leuven, Belgium

Correspondence: r.h.houtkooper@amc.nl; a.w.macinnes@amc.nl

transcription complex and is best known for its role in transcriptional silencing in early embryogenesis by becoming sequestered in the cytoplasm because of phosphorylation by OMA-1 (Oocyte MAturation defective) (Walker et al, 2001). The loss of TAF-4 in *C. elegans* suppresses the lifespan extension phenotype induced by *clk-1* mutations; however, the mechanism through which this happens is not known.

Reducing protein synthesis is another well-known means by which eukaryotes extend the lifespan (Hansen et al, 2007; Pan et al, 2007). Caloric restriction, amino acid reduction, and knocking down myriad factors involved in mRNA translation (such as ribosomal proteins, elongation/initiation factors, or tRNA synthetases) all increase the lifespan (Masoro, 2000; Min & Tatar, 2006; Hansen et al, 2007; Pan et al, 2007). The genetic or pharmacological inhibition of the mechanistic target of rapamycin (mTOR)/nutrient sensing pathway or insulin receptor signaling pathway is also marked by reduced mRNA translation rates and a suppression of polyribosome formation (Genolet et al, 2008; Stout et al, 2013). Proteomic analysis of the *daf-2* mutants previously revealed a substantial reduction of ribosomal proteins, translation factors, and protein metabolism components coupled to a repression of polyribosome formation (Stout et al, 2013). Moreover, our previous work demonstrated that similarly to *daf-2* mutants, mutations in *clk-1* worms also result in a strong repression of polyribosome formation (Essers et al, 2015). This led us to hypothesize the existence of unexplored regulatory links between dysfunctional mitochondria and cytoplasmic protein synthesis that are contributing to eukaryotic lifespan extension. In particular, we investigated which subsets of mRNAs are preferentially translated in the longer lived *C. elegans* with impaired UQ biosynthesis.

# Results

### RNAseq of *clk-1(qm30)* strains shows major changes in transcription

To uncover how the transcriptome of *C. elegans* changes on deletion of *clk-1* and the influence of the *clk-1* mitochondrial localization signal, we performed next-generation sequencing of total RNAs (depleted of ribosomal RNA [rRNA]) isolated from worms at the L4 stage that precedes young adulthood. Total RNA pools depleted of rRNA isolated from *clk-1(qm30)* mutants were compared with (1) *clk-1(qm30)*[+WT] mutants rescued with the WT *clk-1* and (2) *clk-1(qm30)*[+nuc] mutants with predominantly the nuclear form of *clk-1* (Fig 1A) that was described in Monaghan et al (2015). The *clk-1(qm30)*[+WT] mutants have WT phenotypes and are rescued with fully functioning *clk-1*, whereas the *clk-1(qm30)*[+nuc] mutants have *clk-1* only residing in the nucleus and not in the mitochondria (Monaghan et al, 2015). The principal component analysis (PCA) plot and correlation matrix show the expected clustering of the biological triplicates of each strain (Fig 1B–C). Major changes were observed in the RNA pool of the *clk-1(qm30)*[+WT] mutants compared with either *clk-1(qm30)* or *clk-1(qm30)*[+nuc], with 9,626 and 8,685 genes being differentially expressed, respectively. Far fewer differentially expressed genes were measured when comparing *clk-1(qm30)* with *clk-1(qm30)*[+nuc], with only 247 genes up-regulated and 571 genes down-regulated. The subtle changes in RNA expression are consistent with the observation that introducing the nuclear form of

*clk-1* in *clk-1(qm30)* results in only small phenotypic changes compared with the *clk-1(qm30)* mutants (Monaghan et al, 2015; Liu et al, 2017).

### Transcripts encoding the translation machinery are strongly reduced in *clk-1(qm30)*

We next identified the biological pathways that are altered in *clk-1(qm30)* worms using The Database for Annotation, Visualization and Integrated Discovery (DAVID) and Gene Ontology (GO) analysis (Huang da et al, 2009a; Huang da et al, 2009b). Among the most up-regulated processes in *clk-1(qm30)* are metabolic pathways (e.g., oxidation–reduction pathway) and ion transport (Fig 1D). A substantial proportion (25%) of significantly down-regulated processes in *clk-1(qm30)* mutants is involved in reproduction and development (Fig 1D). Developmental delay and reduced progeny have been extensively reported as linked to the longevity phenotype of *clk-1(qm30)* mutants (Larsen et al, 1995; Tissenbaum & Ruvkun, 1998). Remarkably, another 20% of down-regulated GO terms are involved in translation and mRNA processing (Fig 1D). In total, 38 genes encoding ribosomal proteins, initiation, and elongation factors are down-regulated in the *clk-1(qm30)* mutants compared with *clk-1(qm30)*[+WT]. In addition, 28 genes specifically involved in biogenesis and processing of ribosomes were down-regulated in the *clk-1(qm30)* worms. Similar down-regulation of ribosomal proteins and initiation/elongation factors was observed in the *clk-1(qm30)*[+nuc] mutants. These observations suggest that mRNA translation is repressed in the *clk-1(qm30)/*[+nuc] mutants and are in line with the repressed polysome profiles we observed before (Essers et al, 2015).

### There is a minor role for nuclear *clk-1* in transcriptional changes

Because the nuclear form of *clk-1* was suggested to associate with the chromatin in the nucleus (Monaghan et al, 2015), we expected changes in the transcriptome in the mutant expressing exclusively the nuclear form of *clk-1*. However, differentially expressed genes in *clk-1(qm30)*[+nuc] were very similar to the differentially expressed genes *clk-1(qm30)* (Fig 2A). There were 3,720 genes similarly up-regulated in the *clk-1(qm30)*[+nuc] as in the total *clk-1(qm30)* mutants compared with *clk-1(qm30)*[+WT] and 4,397 genes similarly down-regulated in both strains compared with the WT control (Fig 2B). The 292 up-regulated genes and 275 down-regulated genes in the *clk-1(qm30)*[+nuc] strain that were not up- or down-regulated in the *clk-1(qm30)* strain did not lead to significantly clustered GO terms. The 704 genes up-regulated in specifically the *clk-1(qm30)* strain and not in the *clk-1(qm30)*[+nuc] strain were involved in biological processes such as ion transport and oxidation–reduction (Fig 2C). The 805 genes down-regulated in the *clk-1(qm30)* strain but not in *clk-1(qm30)*[+nuc] were involved in biological processes such as reproduction and response to DNA damage stimuli (Fig 2C). These processes also appeared when comparing the *clk-1(qm30)* mutants with WT worms.

### Translation rates are reduced in *clk-1(qm30)* worms

To provide a snapshot of the translational status in the *clk-1(qm30)* strains, we performed polysome profiling of *C. elegans* at the L4 stage. Polysome profiling fractionates total cell lysate over a sucrose gradient by density and enables the distinct measurement of 40S small ribosomal subunits, 60S large ribosomal subunits, 80S

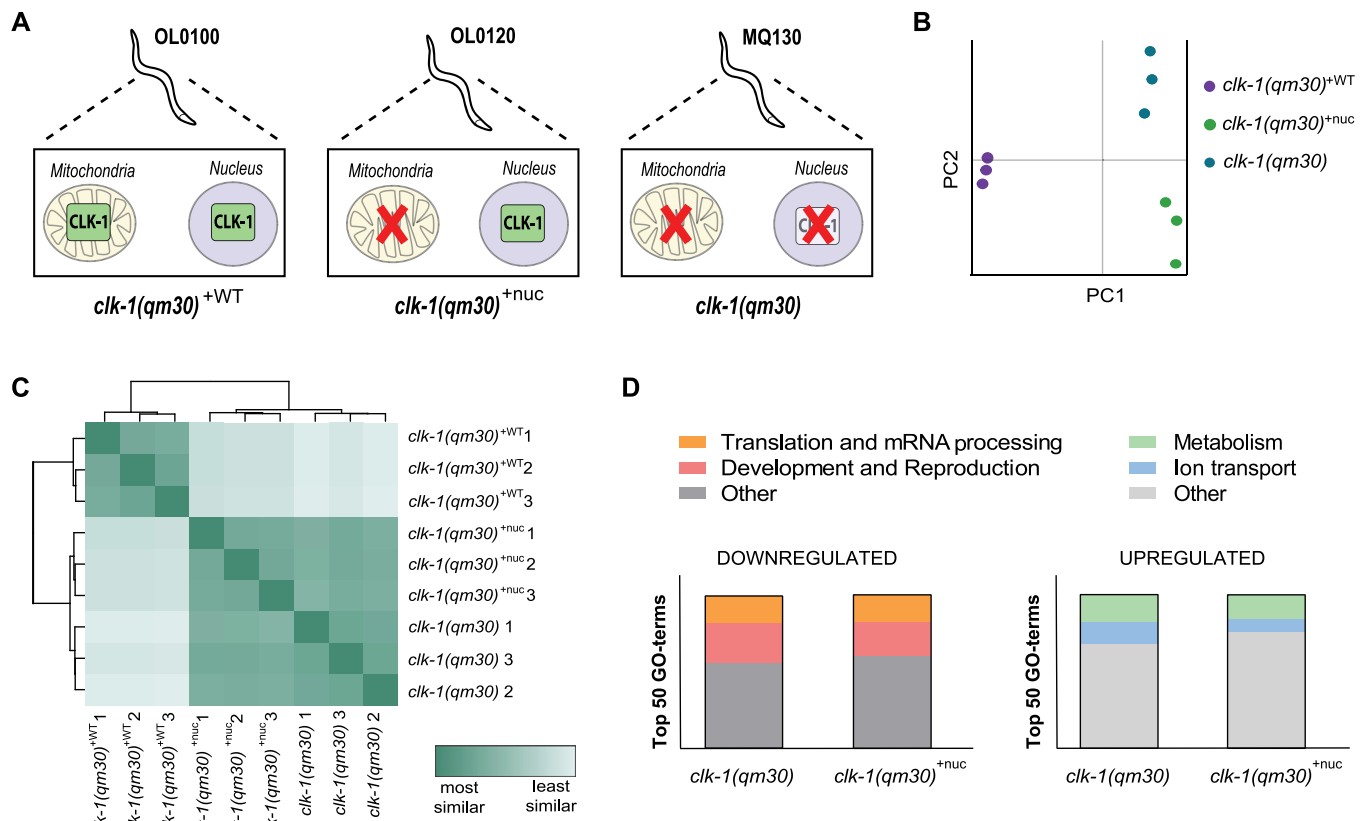

**Figure 1. RNA-seq of the *clk-1(qm30)* (± nuclear or WT *clk-1*) mutants.**
**(A)** Graphical illustration of *clk-1(qm30)*, *clk-1(qm30)*+nuc, and *clk-1(qm30)*+WT lines used for RNA-seq. mRNA was isolated from three biological replicates of each mutant, harvested at the L4 stage. **(B)** PCA of RNA libraries showing clear distinction between *clk-1(qm30)*, *clk-1(qm30)*+nuc, and *clk-1(qm30)*+WT (N = 3 biological replicates for each strain). **(C)** Correlation matrix of RNA-seq samples shows the expected clustering of the biological triplicates of each strain. **(D)** Top 50 down-regulated GO terms (left graph) and up-regulated GO terms (right graph) in *clk-1(qm30)* versus *clk-1(qm30)*+WT (left column) and *clk-1(qm30)*+nuc versus *clk-1(qm30)*+WT (right column). Proportions of GO terms associated with translation and mRNA processing are depicted in yellow, development and reproduction in red, metabolism in green, and ion transport in blue.

monosomes (an mRNA molecule with a single ribosome attached), and multiple polysomes (mRNA molecules with multiple ribosomes attached). Polysome profiles of *clk-1(qm30)* reveal a strong repression of both monosomal and polysomal peaks compared with *clk-1(qm30)*+WT, suggesting a global repression of mRNA translation (Fig 3). Again, the *clk-1(qm30)*+nuc mutants show similar yet slightly less dramatic differences compared with WT versus the total *clk-1(qm30)* mutant compared with WT (Fig 3). Because the *clk-1(qm30)*+nuc does not show striking differences in transcription compared with the total *clk-1(qm30)* mutant, we focused our remaining studies on comparing *clk-1(qm30)* with *clk-1(qm30)*+WT. The polysome profiles, together with the RNAseq results, show that *clk-1(qm30)* mutants are marked by repressed mRNA translation. Further validation of translational repression in *clk-1(qm30)* mutants is shown by quantitative PCR (qPCR) analysis, demonstrating that several mRNAs encoding initiation factors and ribosomal proteins are down-regulated in *clk-1(qm30)* mutants compared with WT worms (Fig S1).

## Total RNA does not mirror highly translated polysomal RNA

In addition to providing a global overview of mRNA translation rates, polysome profiling allows the specific isolation of the RNAs

associated with the polyribosomes for comparison with the total RNAs in different *clk-1(qm30)* strains (Fig 4A). The PCA plot and correlation matrix show clustering of polysomal and total RNA of the biological triplicates of each strain (Fig S2). We observed that changes in the polysomal RNA of *clk-1(qm30)* compared with the polysomal RNA of *clk-1(qm30)*+WT were not simply mirroring the changes observed in the total mRNA pools of these mutants (Fig 4B). For instance, there are 434 genes specifically up-regulated in the polysomal RNA pool carrying highly translated messages that are not up-regulated in the total RNA pool of *clk-1(qm30)* mutants (Fig 4C). A larger set of 1,657 genes is up-regulated in both polysomal and total RNA, whereas the largest part (2,767 genes) is exclusively up-regulated in the total RNA pool (Fig 4C). A similar pattern is observed with down-regulated genes in *clk-1(qm30)* mutants, with 466 genes detected in the polysomal RNA pool, 2,574 in both polysomal and total RNA pools, and 2,628 in the strictly total RNA pool (Fig 4C).

## OXPHOS transcripts are highly enriched in polysomal RNA of *clk-1 (qm30)* strains

To determine the functional relevance of the differentially expressed RNAs in total and polysomal RNA pools, we performed

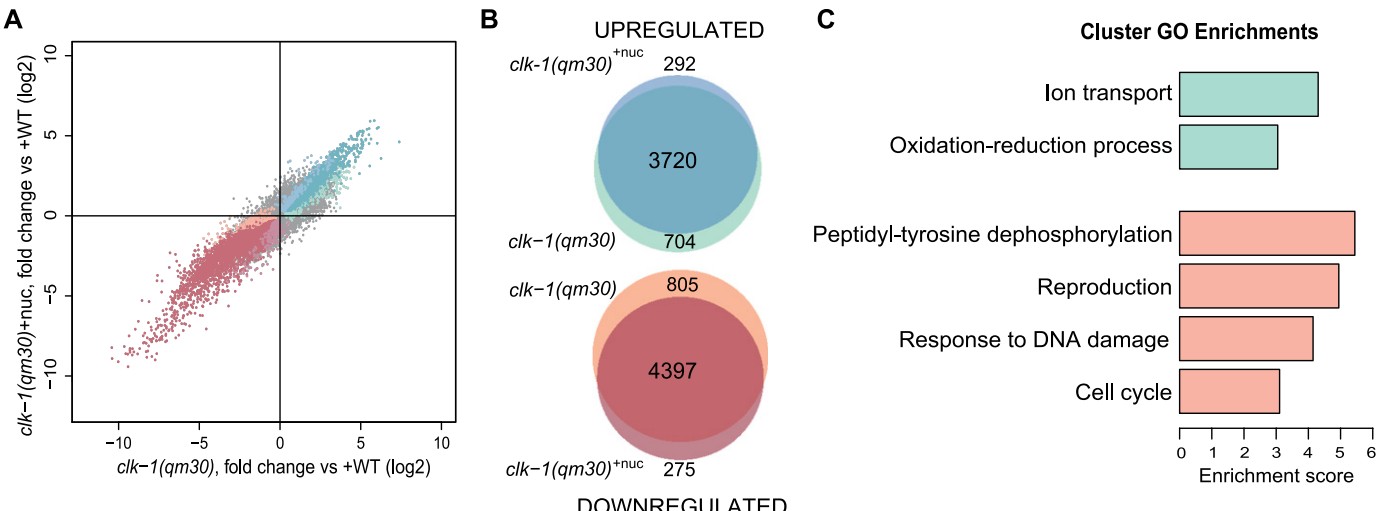

**Figure 2. Differentially expressed genes in *clk-1(qm30)*[+nuc] compared with *clk-1(qm30)*.**
**(A)** Fold changes of differentially expressed genes in *clk-1(qm30)* versus *clk-1(qm30)*[+WT] plotted against fold changes of *clk-1(qm30)*[+nuc] versus *clk-1(qm30)*[+WT] shows a minor role for nuclear *clk-1* in transcriptional changes. Colors of the individual data points correspond to the colors of the groups of genes in the Venn diagram in (B). **(B)** Venn diagram shows that most genes are similarly up-regulated (3720) and down-regulated (4397) in the *clk-1(qm30)*[+nuc] and *clk-1(qm30)* worms when comparing with *clk-1*[+WT]. There were 292 genes exclusively up-regulated in *clk-1(qm30)*[+nuc] (blue) and 704 genes exclusively in *clk-1(qm30)* (green) compared with *clk-1*[+WT]. Furthermore, 175 genes were down-regulated exclusively in *clk-1(qm30)*[+nuc] (red) and 805 genes down-regulated exclusively in *clk-1(qm30)* (orange) compared with *clk-1*[+WT]. Differential expressions in (A) and (B) are with a threshold-adjusted *P* value < 0.01. **(C)** Significant Cluster GO Enrichments (threshold Enrichment Score > 3) associated with the 704 genes specifically up-regulated (green) and 805 down-regulated (orange) in *clk-1(qm30)* strain. There were no significant Cluster GO Enrichments for genes exclusively up or down-regulated in *clk-1(qm30)*[+nuc].

DAVID GO analysis. For instance, ion transport processes are up-regulated in the total mRNA of *clk-1(qm30)* mutants. However, the polysomal expression data indicate that despite this higher abundance in the total pool, mRNAs coding for ion transport are not more highly translated in *clk-1(qm30)* mutants (Fig 4D). In contrast, mRNAs involved in metabolic processes such as oxidation–

reduction, glycolysis, and lipid metabolism are substantially up-regulated in both the total and the polysomal RNA pools (Fig 4D). This suggests a metabolic compensation for the reduction of OXPHOS capacity by glycolysis and lipid metabolism in the *clk-1 (qm30)* mutants. Processes that were exclusively up-regulated in the polysomal RNA pool (and thus exclusively highly translated) in

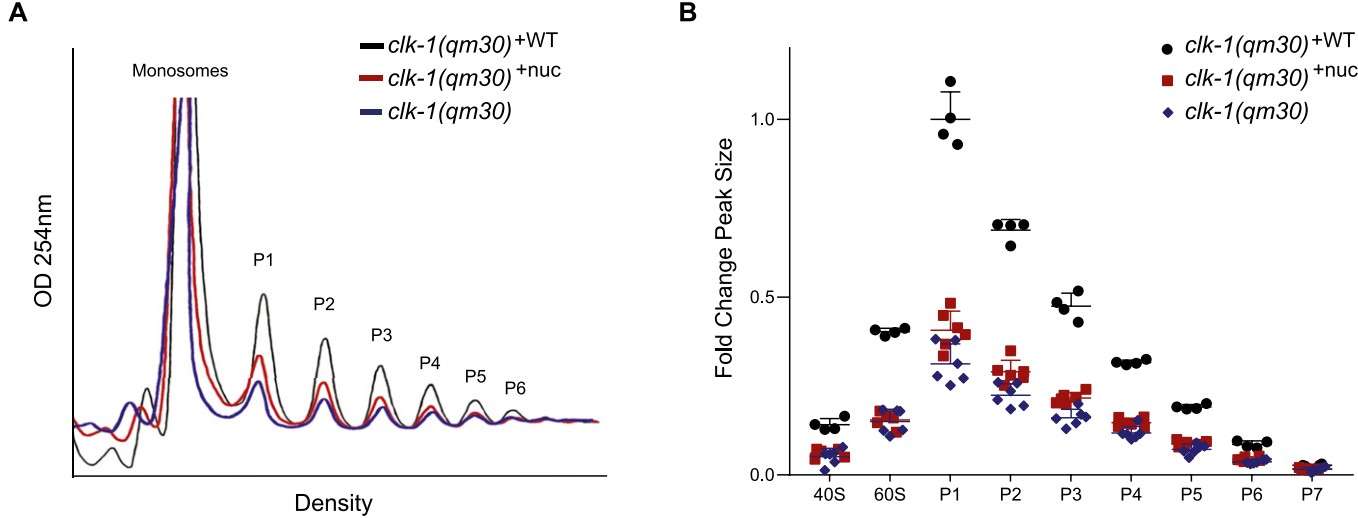

**Figure 3. Repressed polysome profiles in the *clk-1(qm30)* ([±nuc]) mutants.**
**(A)** Representative traces of polysome profiles of *clk-1(qm30)* strains harvested at the L4 stage, when lysate is normalized to total protein levels of 500 μg. The monosomal peak and polysomal peaks (P1–P6) are indicated. **(B)** Quantification of polysome peak sizes (AUC). The fold change is represented compared with P1 of the *clk-1(qm30)*[+WT]. All peaks of *clk-1(qm30)*[+WT] are significantly different from both *clk-1(qm30)*[+nuc] and *clk-1(qm30)*. No peaks were significantly different between *clk-1(qm30)* and *clk-1(qm30)*[+nuc]. Error bars represent mean ± SD. Significance was tested with *t* test and *P*-values were adjusted to correct for multiple testing using the Holm–Sidak method, with α = 0.05.

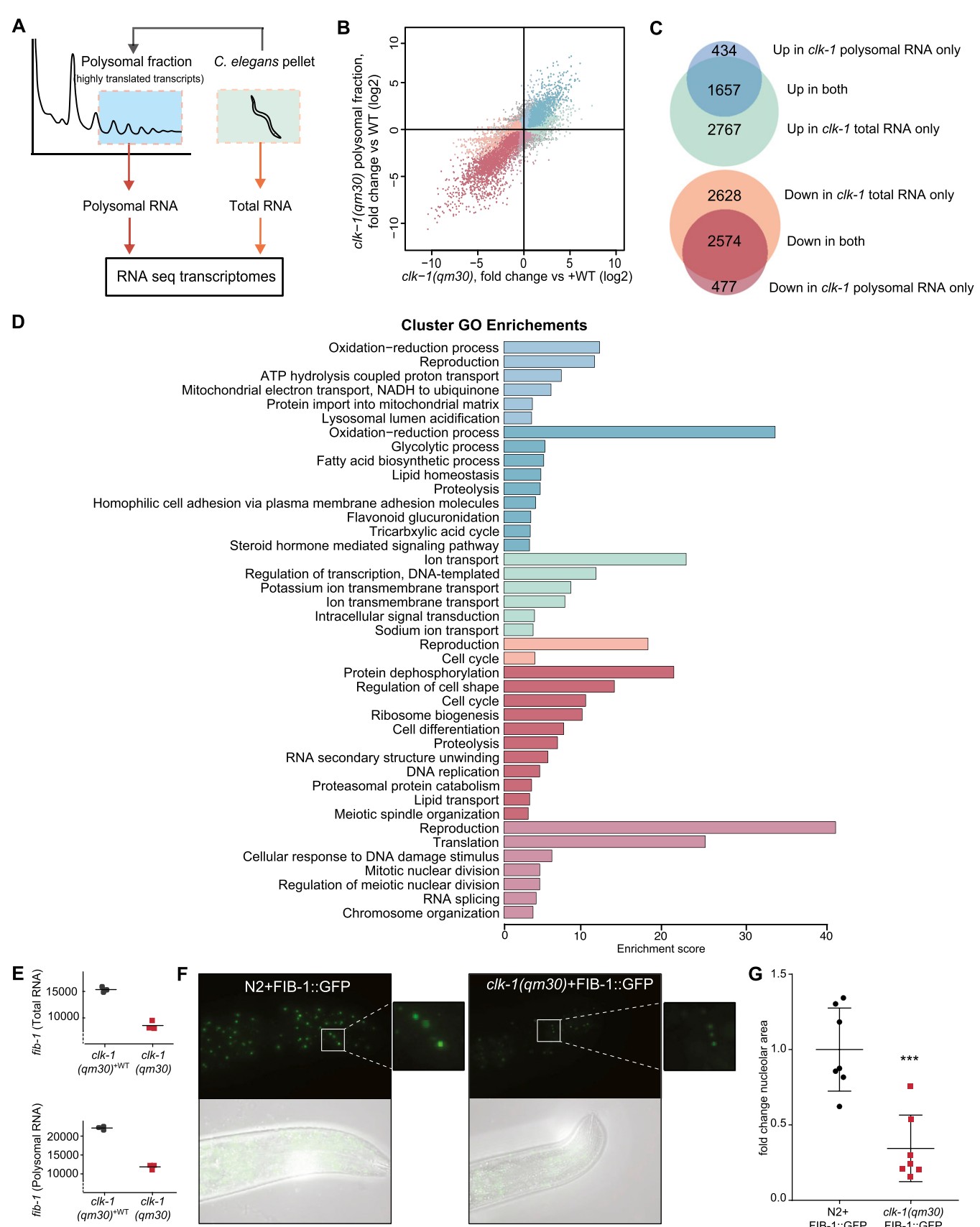

*clk-1(qm30)* mutants are involved in ATP hydrolysis, protein import into mitochondria, and mitochondrial electron transport (Fig 4D). More specifically, 50 transcripts encoding proteins involved in mitochondrial OXPHOS were enriched in *clk-1(qm30)* polysomes (Table 1). This up-regulation does not seem complex-specific because mRNAs coding for proteins of all OXPHOS complexes (I–V) were enriched in *clk-1(qm30)* polysomes. These data suggest a model whereby proteins involved in ATP production and the import of these proteins into the mitochondria are highly translated despite lower overall expression levels in *clk-1(qm30)* mutants compared with WT to attempt to overcome their energy deficit.

### RNAs coding for ribosomal proteins are substantially reduced in *clk-1(qm30)* polysomes

An even more substantial down-regulation of transcripts involved in translation and ribosome biogenesis processes was revealed in the polysomal RNA pool of *clk-1(qm30)* deficient worms compared with *clk-1(qm30)*[+WT], confirming the repression of translational machinery proteins (Table 2). In the polysomal RNA pool of *clk-1(qm30)* mutants, 62 ribosomal proteins, 16 elongation/initiation factors, 14 amino-acyl tRNA synthetases, and 10 other proteins involved in mRNA processing were down-regulated (Table 2). These results are reminiscent of the proteomic analysis demonstrating a significant reduction of ribosomal proteins and translation factors in the *daf-2* insulin receptor mutants (Stout et al, 2013). Furthermore, a subset of these transcripts coding for proteins involved in translation (21 ribosomal proteins, 11 amino-acyl tRNA synthetases, and eight other elongation/initiation factors) was even down-regulated in the total pool of RNA. This indicates that the down-regulation of translation machinery is partially regulated in a transcriptional manner. Interestingly, we observed strong down-regulation of *fib-1*, a gene coding for a protein involved in the regulation and maturation of rRNA, in both total and polysomal RNA (Fig 4E). The FIB-1 protein resides in the nucleoli where ribosome biogenesis initiates. It was previously shown that small nucleoli are a cellular hallmark of longevity and metabolic health (Buchwalter & Hetzer, 2017; Tiku et al, 2017). A FIB-1::GFP reporter strain for nucleoli revealed reduced FIB-1 levels in other long-lived *C. elegans* strains (Tiku et al, 2017). In line with this and the RNA-seq data presented here, the *clk-1(qm30)* worms also show reduced FIB-1 levels compared with WT N2 worms that is accompanied by reduced nucleolar size (Fig 4F and G). Taken together, it appears that an attempt to overcome an energy deficit in *clk-1(qm30)* mutants is coupled to the suppression of ribosome biogenesis and mRNA translation, which would conserve flagging energy for other critical cell processes.

### Differential translational efficiency (TE) of mRNA reveals a role for TOR signaling

Because there are mRNAs that are selectively being translated in the *clk-1(qm30)* mutant or the WT independently of their abundance in the total mRNA pool, we calculated the TE of individual mRNA transcripts by taking the ratio between total RNA and polysomal RNA (Fig 5A). DAVID GO analysis on transcripts with high or low TE revealed similar biological processes as observed in the groups of genes specifically up- and down-regulated in the polysomal RNA presented in Fig 4D (Fig 5B). The list of genes with significantly different TEs was then analyzed using the "worm longevity pathway" in Kyoto Encyclopedia of Genes and Genomes (Ogata et al, 1999) using Pathview (Luo & Brouwer, 2013), to investigate the overlap in longevity pathways known in *C. elegans*. Besides hits in the UPR[mt] and stress resistance, both already described in *clk-1(qm30)* mutants (Durieux et al, 2011), the results also revealed possible involvement of the mTOR signaling pathway in the *clk-1(qm30)* mutants. Hence, we also checked the mTOR signaling pathway in Kyoto Encyclopedia of Genes and Genomes Pathview. In *C. elegans*, homologs of TOR pathway genes all extend the lifespan upon siRNA knockdown (Korta et al, 2012). Interestingly, our RNA-seq data show that *let-363* (homologue of TOR) is preferentially translated in WT but not in *clk-1(qm30)* mutants (Fig 5C and D). Other members of the TOR complex 1 (TORC1), including *daf-15* (RAPTOR), *clk-2* (TEL2), and *R10H10.7* (TTI1), also showed decreased TE in *clk-1(qm30)* mutants (Fig 5C and D). Moreover, our results show that a key player in autophagy, *lgg-1*, is preferentially translated in *clk-1(qm30)* mutants but not in the *clk-1(qm30)*[+WT] (Fig 5C and D).

### *taf-4* is required for repressed polysome profiles in *clk-1(qm30)*

TAF-4 is one of few transcription factors reported to be critical for the *clk-1(qm30)* longevity phenotype (Khan et al, 2013). To test if TAF-4 has a role in polysome formation in *clk-1(qm30)* worms, we performed RNAi of *taf-4* in the *clk-1(qm30)* mutants. Here, we show that RNAi of *taf-4* significantly increases the monosomal and polysomal peaks of *clk-1(qm30)* compared with the control empty vector HT115 (Fig 6A and C), suggesting this transcription factor plays a direct role in the suppression of polyribosome formation upon the loss of UQ biosynthesis. This increase is not observed in the polysome profiles upon *taf-4* RNAi in the *clk-1(qm30)*[+WT] (Fig

**Figure 4. Analysis of total and polysomal RNA *clk-1(qm30)* and *clk-1(qm30)*[+WT].**
**(A)** Schematic representation of polysomal fraction (blue) used for RNA isolation and total RNA (green) from *C. elegans* mutants *clk-1(qm30)* and *clk-1(qm30)*[+WT] that were used for RNAseq. **(B)** Fold changes of differentially expressed genes in the total RNA of *clk-1(qm30)* versus *clk-1(qm30)*[+WT] plotted against fold changes in polysomal RNA of *clk-1(qm30)* versus *clk-1(qm30)*[+WT]. Colors of the individual data points correspond to the colors of the groups of genes in the Venn diagram in (C). **(C)** Venn diagram showing 1657 genes were similarly up-regulated and 2574 genes down-regulated in both the polysomal as the total RNA in the *clk-1(qm30)* strain compared with *clk-1(qm30)*[+WT]. A total of 434 genes were exclusively up-regulated in the polysomal RNA (blue), whereas 2767 genes were exclusively up-regulated in the total RNA (green). Furthermore, 477 genes were exclusively down-regulated in the polysomal RNA (red) and 2628 genes were exclusively down-regulated in the total RNA. Differential expressions (B, C) are with a threshold-adjusted *P* value < 0.01. **(D)** Significant Cluster GO Enrichments (threshold Enrichment Score > 3) associated with the genes specifically up- and down-regulated in groups of genes in (C) (colors of the bars correspond again to the color of the groups in (C)). **(E)** Fib-1 is reduced in both the total and polysomal RNA of *clk-1(qm30)* compared with *clk-1(qm30)*[+WT]. **(F, G)** FIB-1 levels are reduced in *clk-1(qm30)* worms compared with N2 worms. Error bars represent mean ± SD, significance was tested with *t* test, ***P < 0.0005.

**Table 1.  Identified polysomal transcripts involved in OXPHOS that, compared with *clk-1(qm30)*<sup>+WT</sup>, are enriched in *clk-1(qm30)*.**

| Accession | Gene name | Description | log2 fold change polysomal RNA |
|---|---|---|---|
| Complex I | | | |
| C16A3.5 | C16A3.5 | Orthologue B9 subunit of the mitochondrial complex I | 0.42 |
| C18E9.4 | C18E9.4 | Orthologue B12 subunit of the mitochondrial complex I | 0.61 |
| C25H3.9 | C25H3.9 | Orthologue B5 subunit of the mitochondrial complex I | 0.40 |
| C33A12.1 | C33A12.1 | Orthologue A5/B13 subunit of the mitochondrial complex I | 0.43 |
| C34B2.8 | C34B2.8 | Orthologue A13 subunit of the mitochondrial complex I | 0.48 |
| F37C12.3 | F37C12.3 | Orthologue AB1 subunit of the mitochondrial complex I | 0.53 |
| F42G8.10 | F42G8.10 | Orthologue B11 subunit of the mitochondrial complex I | 0.38 |
| F44G4.2 | F44G4.2 | Orthologue B2 subunit of the mitochondrial complex I | 0.54 |
| F53F4.10 | F53F4.10 | Orthologue NADH-UQ oxidoreductase flavoprotein 2 | 0.45 |
| ZK973.10 | lpd-5 | NADH-UQ oxidoreductase fe-s protein 4 | 0.37 |
| W10D5.2 | nduf-7 | NADH-UQ oxidoreductase fe-s protein 7 | 0.47 |
| C09H10.3 | nuo-1 | NADH UQ oxidoreductase 1 | 0.35 |
| T10E9.7 | nuo-2 | NADH UQ oxidoreductase 2 | 0.47 |
| W01A8.4 | nuo-6 | NADH UQ oxidoreductase 6 | 0.47 |
| T20H4.5 | T20H4.5 | Orthologue NADH-UQ oxidoreductase Fe-S protein 8 | 0.39 |
| Y53G8AL.2 | Y53G8AL.2 | Orthologue A9 subunit of the mitochondrial complex I | 0.48 |
| Complex II | | | |
| F42A8.2 | sdhb-1 | Succinate dehydrogenase complex subunit B | 0.37 |
| F33A8.5 | sdhd-1 | Succinate dehydrogenase complex subunit D | 0.74 |
| Complex III | | | |
| C54G4.8 | cyc-1 | Cytochrome C reductase | 0.35 |
| F42G8.12 | isp-1 | Rieske iron sulphur protein subunit of the mitochondrial complex III | 0.27 |
| R07E4.3 | R07E4.3 | Orthologue subunit VII ubiquinol–cytochrome c reductase complex III | 0.54 |
| T02H6.11 | T02H6.11 | Ubiquinol–cytochrome c reductase binding protein | 0.40 |
| T27E9.2 | T27E9.2 | Ubiquinol–cytochrome c reductase hinge protein | 0.54 |
| F57B10.14 | ucr-11 | Ubiquinol–cytochrome c oxidoreductase complex | 0.75 |
| Complex IV | | | |
| F26E4.9 | cco-1 | Cytochrome C oxidase | 0.46 |
| Y37D8A.14 | cco-2 | Cytochrome C oxidase | 0.46 |
| F26E4.6 | F26E4.6 | Orthologue cytochrome c oxidase subunit 7C | 0.55 |
| F29C4.2 | F29C4.2 | Orthologue cytochrome c oxidase subunit 6C | 0.66 |
| F54D8.2 | tag-174 | Orthologue cytochrome c oxidase subunit 6A2 | 0.45 |
| Y71H2AM.5 | Y71H2AM.5 | Orthologue cytochrome c oxidase subunit 6B1 | 0.48 |
| Complex V | | | |
| C53B7.4 | asg-2 | ATP synthase G homolog | 0.79 |
| C06H2.1 | atp-5 | ATP synthase subunit | 0.45 |
| F32D1.2 | hpo-18 | Orthologue of ATP synthase, H+ transporting, mitochondrial F1 complex, ε subunit | 0.56 |
| R04F11.2 | R04F11.2 | Orthologue of ATP synthase, H+ transporting, mitochondrial Fo complex, ε subunit | 0.51 |
| R53.4 | R53.4 | Mitochondrial ATP synthase subunit f homolog | 0.34 |

**Table 1. Continued**

| Accession | Gene name | Description | log2 fold change polysomal RNA |
|---|---|---|---|
| R10E11.8 | vha-1 | Vacuolar H ATPase 1 | 0.50 |
| R10E11.2 | vha-2 | Vacuolar H ATPase 2 | 0.63 |
| Y38F2AL.4 | vha-3 | Vacuolar H ATPase 3 | 0.58 |
| T01H3.1 | vha-4 | Vacuolar H ATPase 4 | 0.61 |
| VW02B12L.1 | vha-6 | Vacuolar H ATPase 6 | 0.59 |
| C17H12.14 | vha-8 | Vacuolar H ATPase 8 | 0.66 |
| ZK970.4 | vha-9 | Vacuolar H ATPase 9 | 0.43 |
| F46F11.5 | vha-10 | Vacuolar H ATPase 10 | 0.64 |
| Y38F2AL.3 | vha-11 | Vacuolar H ATPase 11 | 0.58 |
| F20B6.2 | vha-12 | Vacuolar H ATPase 12 | 0.34 |
| Y49A3A.2 | vha-13 | Vacuolar H ATPase 13 | 0.35 |
| F55H2.2 | vha-14 | Vacuolar H ATPase 14 | 0.49 |
| T14F9.1 | vha-15 | Vacuolar H ATPase 15 | 0.51 |
| C30F8.2 | vha-16 | Vacuolar H ATPase 16 | 0.55 |
| Y69A2AR.18 | Y69A2AR.18 | Orthologue of ATP synthase, H+ transporting, mitochondrial F1 complex, γ subunit | 0.25 |

6B). These data suggest that TAF-4 loss partially restores polyribosome formation in *clk-1(qm30)* mutants similar to the previously reported attenuation of the longevity phenotype (Khan et al, 2013). Because TAF-4 is a transcription factor, we looked at changes in expression of specific genes involved in translation upon *taf-4* RNAi in the *clk-1(qm30)* mutants. We observed significant up-regulation of the translation initiation factor *eif-1* when *taf-4* is knocked down in *clk-1(qm30)* worms, suggesting that TAF-4 has a direct role in repressing *eif-1* expression (Fig 6D). *Eif-1* is important for translation initiation, and yeast *eif-1* mutants (*sui1⁻*) show a reduction of polysomes compared with WT (Yoon & Donahue, 1992). Moreover, the reduction of *eif-1* extends lifespan in N2 worms (Wang et al, 2010). Taken all together, these results suggest that the reduction of the translation machinery observed in *clk-1(qm30)* mutants may at least in part be driven by TAF-4 repression, and may help to explain the previously described link between TAF-4– and *clk-1(qm30)*– mediated lifespan extension (Khan et al, 2013).

## Discussion

This study was initiated following our previous observation that the polysome profiles of *clk-1(qm30)* mutants revealed the same repression of polysomes as in the *daf-2* mutants (Essers et al, 2015). We continued this line of inquiry by isolating and sequencing polysomal and total RNAs from *clk-1(qm30)* mutants and compared them with *clk-1(qm30)⁺WT* worms. These results demonstrated that compared with *clk-1(qm30)⁺WT*, very few mRNAs coding for components of the translation machinery such as initiation factors, elongation factors, and ribosomal proteins are present on *clk-1 (qm30)* polyribosomes. In contrast, the *clk-1(qm30)* polyribosomes reveal a strong preference for translating RNAs coding for OXPHOS

proteins. The reduction in the TE of mRNAs coding for the translation machinery is highly reminiscent of the proteomic landscape we described in the *daf-2* mutants, where there is a scarcity of ribosomal proteins (RPs) yet also no appreciable reduction in total RP mRNA expression (Depuydt et al, 2013; Essers et al, 2015). In addition, the small nucleoli in *clk-1(qm30)* worms confirming reduced ribosome biogenesis is a hallmark of cellular aging that was previously observed in other longevity mutants (Tiku et al, 2017). Taken all together, the results suggest that reduced ribosome biogenesis and shutting down protein synthesis by altering the TE of mRNAs coding for the translation machinery is a common mechanism used by several different systems that drive lifespan extension.

Our results go on to suggest that the proposed nuclear form of CLK-1 has a relatively small effect on alleviating the polyribosome reduction observed in the *clk-1(qm30)* mutants. The subtle differences in lifespan and development previously described (Monaghan et al, 2015) might be explained by reestablishing expression of some reproduction genes in the *clk-1(qm30)⁺nuc* mutant, which are well known to associate with lifespan (Hsin & Kenyon, 1999; Monaghan et al, 2015). Because it has been shown that under certain conditions proteins may be imported into the mitochondria without any MTS (Ruan et al, 2017), it is possible that a small proportion of the nuclear CLK-1 enters the mitochondria without an MTS. The present study does not disprove the existence of nuclear form of CLK-1; however, an obvious function of a nuclear CLK-1 is not apparent with these data.

The inhibition of respiration in the *clk-1(qm30)* mutants triggers a retrograde response by up-regulating cell-protective and metabolic genes (Cristina et al, 2009). Our work is in line with these findings and in this study, we confirm the importance of retrograde communication between stressed mitochondria and the rest of the cell. In *Drosophila*, it was shown that dietary restriction–induced longevity was regulated by activation of the translational repressor

**Table 2.  Identified polysomal transcripts involved in mRNA translation that, compared with *clk-1(qm30)*⁺ᵂᵀ, are reduced in *clk-1(qm30)*.**

| Accession | Gene name | Description | log2 fold change |
|---|---|---|---|
| Translation | | | |
| ZC434.5 | ears-1 | Glutamyl(E) amino-acyl tRNA synthetase | −0.51 |
| F31E3.5 | eef-1A.1 | Eukaryotic translation elongation factor 1-α | −0.67 |
| F25H5.4 | eef-2 | Eukaryotic translation elongation factor 2 | −0.53 |
| C27D11.1 | egl-45 | Eukaryotic translation initiation factor 3 subunit A | −0.50 |
| F11A3.2 | eif-2Bdelta | Eukaryotic translation initiation factor 2B subunit Δ | −0.79 |
| Y54E2A.11 | eif-3.B | Eukaryotic translation initiation factor 3 subunit B | −0.38 |
| T23D8.4 | eif-3.C | Eukaryotic translation initiation factor 3 subunit C | −0.43 |
| R11A8.6 | iars-1 | Isoleucyl(I) amino-acyl tRNA synthetase | −0.26 |
| M110.4 | ifg-1 | Initiation factor 4G (eIF4G) family | −0.43 |
| K10C3.5 | K10C3.5 | Orthologue of Ria1p | −0.92 |
| R74.1 | lars-1 | Leucyl(L) amino-acyl tRNA synthetase | −0.36 |
| C47E12.1 | sars-1 | Seryl(S) amino-acyl tRNA synthetase | −0.43 |
| F28H1.3 | aars-2 | Alanyl(A) amino-acyl tRNA synthetase | −0.46 |
| K08F11.3 | cif-1 | COP9/signalosome and eIF3 complex shared subunit | −0.37 |
| Y41E3.10 | eef-1B.2 | Eukaryotic translation elongation factor | −0.31 |
| Y37E3.10 | eif-2A | Eukaryotic initiation factor | −1.06 |
| R08D7.3 | eif-3.D | Eukaryotic initiation factor | −1.61 |
| Y40B1B.5 | eif-3.J | Eukaryotic initiation factor | −1.24 |
| C17G10.9 | eif-3.L | Eukaryotic initiation factor | −0.61 |
| C47B2.5 | eif-6 | Eukaryotic initiation factor | −0.53 |
| T05H4.6 | erfa-1 | Eukaryotic release factor homolog | −0.30 |
| F22B5.9 | fars-3 | Phenylalanyl(F) amino-acyl tRNA synthetase | −1.65 |
| T10F2.1 | gars-1 | Glycyl(G) amino-acyl tRNA synthetase | −0.41 |
| F53A2.6 | ife-1 | Initiation factor 4E (eIF4E) family | −0.72 |
| B0348.6 | ife-3 | Initiation factor 4E (eIF4E) family | −0.55 |
| T05G5.10 | iff-1 | Initiation factor five (eIF-5A) homolog | −0.31 |
| Y54F10BM.2 | iffb-1 | Initiation factor five B (eIF5B) | −0.75 |
| F57B9.6 | inf-1 | Initiation factor | −0.31 |
| T02G5.9 | kars-1 | Lysyl(K) amino-acyl tRNA synthetase | −0.30 |
| F58B3.5 | mars-1 | Methionyl(M) amino-acyl tRNA synthetase | −0.62 |
| F22D6.3 | nars-1 | Asparaginyl(N) amino-acyl tRNA synthetase | −0.64 |
| Y41E3.4 | qars-1 | Glutaminyl(Q) amino-acyl tRNA synthetase | −0.43 |
| F26F4.10 | rars-1 | Arginyl(R) amino-acyl tRNA synthetase | −0.55 |
| C47D12.6 | tars-1 | Threonyl(T) amino-acyl tRNA synthetase | −0.44 |
| Y87G2A.5 | vrs-2 | Valyl(V) amino-acyl tRNA synthetase | −0.41 |
| Y80D3A.1 | wars-1 | Tryptophanyl(W) amino-acyl tRNA synthetase | −0.67 |
| Y105E8A.19 | yars-1 | Tyrosinyl(Y) amino-acyl tRNA synthetase | −0.90 |
| Ribosome | | | |
| Y71F9AL.13 | rpl-1 | Large ribosomal subunit L1 protein | −0.53 |
| B0250.1 | rpl-2 | Large ribosomal subunit L2 protein | −0.82 |
| F13B10.2 | rpl-3 | Large ribosomal subunit L3 protein | −0.75 |
| B0041.4 | rpl-4 | Large ribosomal subunit L4 protein | −0.95 |

*Clk-1* and mRNA translation   Molenaars et al.   https://doi.org/10.26508/lsa.201800082   vol 1 | no 5 | e201800082   **9 of 17**

| Accession | Gene name | Description | log2 fold change |
|---|---|---|---|
| 54C9.5 | rpl-5 | Large ribosomal subunit L5 protein | −0.58 |
| R151.3 | rpl-6 | Large ribosomal subunit L6 protein | −0.37 |
| Y24D9A.4 | rpl-7A | Large ribosomal subunit L7A protein | −0.44 |
| R13A5.8 | rpl-9 | Large ribosomal subunit L9 protein | −0.67 |
| JC8.3 | rpl-12 | Large ribosomal subunit L12 protein | −0.48 |
| C32E8.2 | rpl-13 | Large ribosomal subunit L13 protein | −0.56 |
| C04F12.4 | rpl-14 | Large ribosomal subunit L14 protein | −0.30 |
| K11H12.2 | rpl-15 | Large ribosomal subunit L15 protein | −0.61 |
| M01F1.2 | rpl-16 | Large ribosomal subunit L16 protein | −0.73 |
| Y48G8AL.8 | rpl-17 | Large ribosomal subunit L17 protein | −0.45 |
| Y45F10D.12 | rpl-18 | Large ribosomal subunit L18 protein | −0.66 |
| C09D4.5 | rpl-19 | Large ribosomal subunit L18A protein | −0.51 |
| E04A4.8 | rpl-20 | Large ribosomal subunit L20 protein | −0.69 |
| C14B9.7 | rpl-21 | Large ribosomal subunit L18A protein | −0.51 |
| D1007.12 | rpl-24.1 | Large ribosomal subunit L20 protein | −0.51 |
| F28C6.7 | rpl-26 | Large ribosomal subunit L18A protein | −0.29 |
| T24B8.1 | rpl-32 | Large ribosomal subunit L20 protein | −0.26 |
| F37C12.4 | rpl-36 | Large ribosomal subunit L18A protein | −0.39 |
| C26F1.9 | rpl-39 | Large ribosomal subunit L20 protein | −0.98 |
| C09H10.2 | rpl-41 | Large ribosomal subunit L18A protein | −0.41 |
| Y48B6A.2 | rpl-43 | Large ribosomal subunit L20 protein | −0.50 |
| B0393.1 | rps-0 | Small ribosomal subunit S protein | −0.49 |
| F56F3.5 | rps-1 | Small ribosomal subunit S1 protein | −0.57 |
| C49H3.11 | rps-2 | Small ribosomal subunit S2 protein | −0.50 |
| C23G10.3 | rps-3 | Small ribosomal subunit S3 protein | −0.50 |
| Y43B11AR.4 | rps-4 | Small ribosomal subunit S4 protein | −0.54 |
| T05E11.1 | rps-5 | Small ribosomal subunit S5 protein | −0.45 |
| Y71A12B.1 | rps-6 | Small ribosomal subunit S6 protein | −0.83 |
| ZC434.2 | rps-7 | Small ribosomal subunit S7 protein | −0.39 |
| F40F11.1 | rps-11 | Small ribosomal subunit S11 protein | −0.57 |
| F37C12.9 | rps-14 | Small ribosomal subunit S14 protein | −0.51 |
| T08B2.10 | rps-17 | Small ribosomal subunit S17 protein | −0.41 |
| Y57G11C.16 | rps-18 | Small ribosomal subunit S18 protein | −0.48 |
| F37C12.11 | rps-21 | Small ribosomal subunit S21 protein | −0.35 |
| F53A3.3 | rps-22 | Small ribosomal subunit S22 protein | −0.44 |
| T07A9.11 | rps-24 | Small ribosomal subunit S24 protein | −0.53 |
| H06I04.4 | ubl-1 | Small ribosomal subunit S27a protein | −0.37 |
| Y62E10A.1 | rla-2 | Ribosomal protein, large subunit, acidic (P1) | −0.39 |
| F10B5.1 | rpl-10 | Large ribosomal subunit L10 protein | −0.42 |
| T22F3.4 | rpl-11.1 | Large ribosomal subunit L11 protein | −1.26 |
| C27A2.2 | rpl-22 | Large ribosomal subunit L22 protein | −0.47 |
| C03D6.8 | rpl-24.2 | Large ribosomal subunit L24 protein | −0.68 |
| F52B5.6 | rpl-25.2 | Large ribosomal subunit L23a protein | −0.82 |

**Table 2. Continued**

| Accession | Gene name | Description | log2 fold change |
|---|---|---|---|
| C53H9.1 | rpl-27 | Large ribosomal subunit L27 protein | −0.52 |
| R11D1.8 | rpl-28 | Large ribosomal subunit L28 protein | −0.65 |
| C42C1.14 | rpl-34 | Large ribosomal subunit L34 protein | −0.53 |
| ZK1010.1 | rpl-40 | Large ribosomal subunit L40 protein | −0.52 |
| C16A3.9 | rps-13 | Small ribosomal subunit S13 protein | −0.57 |
| F36A2.6 | rps-15 | Small ribosomal subunit S15 protein | −0.49 |
| T05F1.3 | rps-19 | Small ribosomal subunit S19 protein | −0.44 |
| Y105E8A.16 | rps-20 | Small ribosomal subunit S20 protein | −0.47 |
| F28D1.7 | rps-23 | Small ribosomal subunit S23 protein | −0.62 |
| Y41D4B.5 | rps-28 | Small ribosomal subunit S28 protein | −0.64 |
| C26F1.4 | rps-30 | Small ribosomal subunit S30 protein and ubiquitin | −0.43 |
| F42C5.8 | rps-8 | Small ribosomal subunit S8 protein | −0.70 |
| F40F8.10 | rps-9 | Small ribosomal subunit S9 protein | −0.41 |
| W01D2.1 | W01D2.1 | Orthologue of human RPL37 (ribosomal protein L37) | −0.41 |
| Y37E3.8 | Y37E3.8 | Orthologue of human RPL27A (ribosomal protein L27a) | −0.40 |
| mRNA processing | | | |
| W03H9.4 | cacn-1 | CACtiN (*Drosophila* cactus interacting protein) homolog | −0.65 |
| F32B6.3 | F32B6.3 | Pre-mRNA processing factor 18 | −0.85 |
| K07C5.6 | K07C5.6 | Homologue of splicing factor SLU7 | −0.95 |
| F33A8.1 | let-858 | Similarity to eukaryotic initiation factor eIF-4 γ | −0.47 |
| C04H5.6 | mog-4 | DEAH helicase | −0.65 |
| EEED8.5 | mog-5 | DEAH box helicase 8 | −0.52 |
| C50C3.6 | prp-8 | Yeast PRP (splicing factor) related | −0.27 |
| Y46G5A.4 | snrp-200 | Small nuclear ribonucleoprotein homologue | −0.38 |
| C07E3.1 | stip-1 | Septin- and tuftelin-interacting protein homologue | −0.84 |
| W04D2.6 | W04D2.6 | Orthologue of human RBM25 | −0.57 |

Highlighted in grey are the transcripts that were up-regulated in both the polysomal RNA as the total pool of RNA.

4E-BP, which is the eukaryotic translation initiation factor 4E binding protein (Zid et al, 2009). This resulted in increased ribosomal loading of nuclear-encoded OXPHOS mRNAs. This high TE of mitochondrial proteins was crucial for the dietary restriction–induced lifespan extension because inhibition of individual mitochondrial subunits of OXPHOS complexes diminished the lifespan extension (Zid et al, 2009). Another study showed that reduction of cytosolic protein synthesis could suppress ageing-related mitochondrial degeneration in yeast (Wang et al, 2008). This postponed mitochondrial degeneration upon reducing cytoplasmic protein synthesis together with our results suggesting a robust strategy where mitochondria communicate with the cytoplasmic translation machinery to equilibrate cellular energy generated by one compartment and used in another. Interestingly, the mitochondrial dysfunction observed in young *Mclk1*$^{+/−}$ mice was shown to paradoxically result in an almost complete protection from the age-dependent loss of mitochondrial function later in life (Lapointe et al, 2009). The increased TE of mitochondrial mRNAs we observed here in the mitochondrial *clk-1(qm30)* mutants could induce this physiological state that ultimately develops long-term beneficial effects for aging.

Our results found that mRNAs coding for OXPHOS complexes I–V substantially enriched on polysomes of *clk-1(qm30)* worms. Given the marked reduction of total polysomes in these mutants, we can assume that most of the few remaining polysomes are present solely to maintain OXPHOS complexes at viable levels. This translational up-regulation of OXPHOS mRNAs on ribosomes was also recently observed by ribosome profiling in the context of host shutoff during vaccinia virus infection in human cells (Dai et al, 2017). This host shutoff facilitates resource availability for viral evasion and is coupled to a global inhibition of protein synthesis in the host (Bercovich-Kinori et al, 2016). In addition, the authors propose that this translational up-regulation of OXPHOS transcripts is due to their relatively short 5′ UTRs, which are known to play a regulatory role in TE (Araujo et al, 2012; Dai et al, 2017). A similar mechanism coupling energy metabolism and translational control is observed in the present study in the context of longevity.

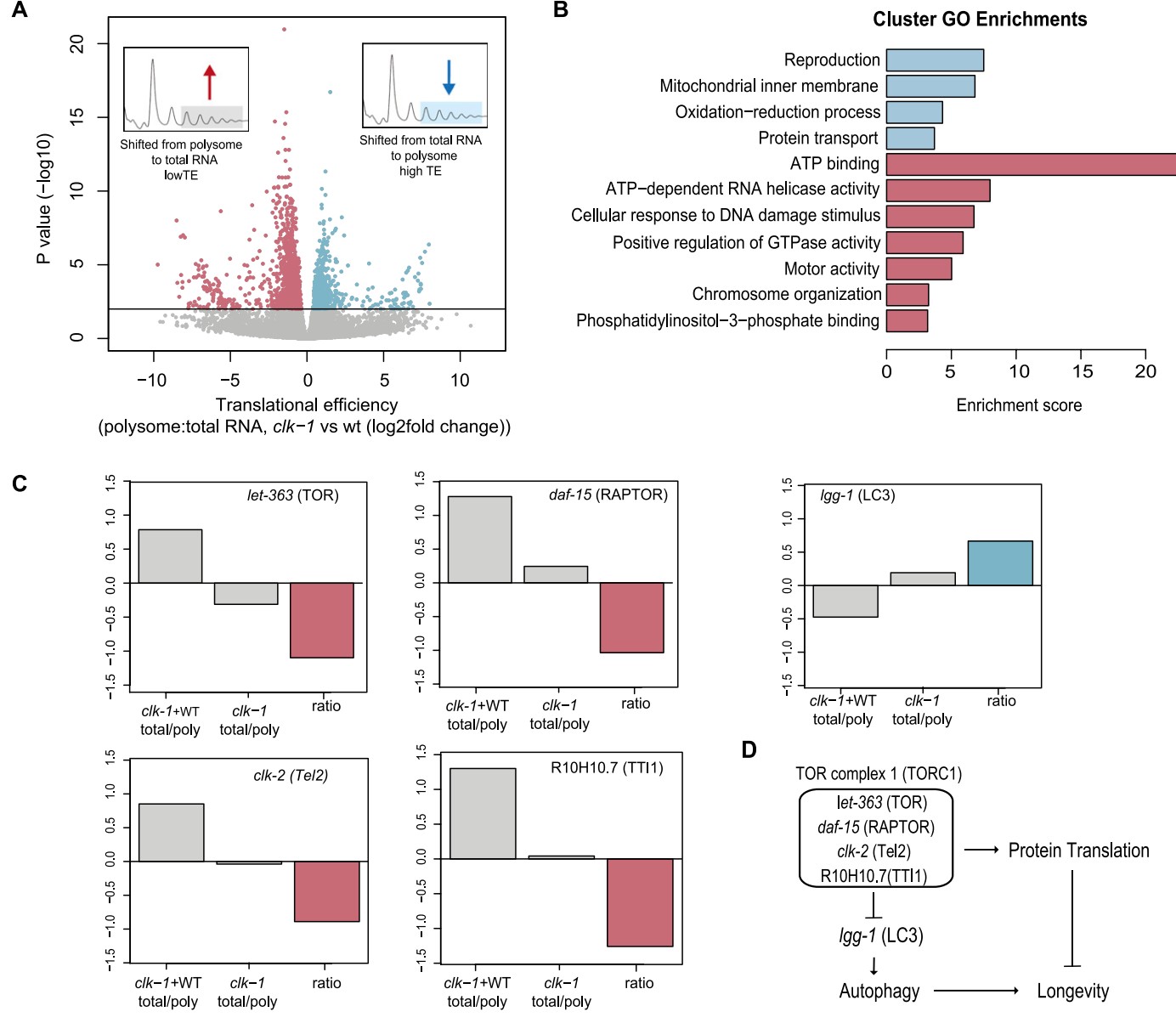

**Figure 5. Translation efficiency (TE) of transcripts in *clk-1(qm30)* worms.**
**(A)** Volcano plot of log2 fold change of TE (total RNA:polysomal RNA) of transcripts *clk-1(qm30):clk-1(qm30)*[*WT]. The blue data points represent transcripts with high TE in *clk-1(qm30)* worms being shifted from the total RNA to the highly translated polysomal RNA. The red data points represent transcripts that have low TE in *clk-1(qm30)* being shifted from polysomal RNA to the total RNA. Differentially translationally regulated genes in red and blue with threshold-adjusted *P* value < 0.01. **(B)** Significant Cluster GO Enrichments (threshold Enrichment Score > 3) associated with significantly different TEs (colors of the bars correspond again to the data points in (A)). **(C)** TE of individual transcripts involved in TOR pathway in *clk-1(qm30)*[*WT] (left bar) *clk-1(qm30)* (middle bar) and their ratio (right bar). Colors of the ratio bars correspond again to the data points in (A). *Clk-1(qm30)* worms slow reduced TE of *let-363*, *daf-15*, *clk-2*, and R10H10.7 and an increased TE for *lgg-1*. **(D)** Schematic overview of transcripts involved in TOR pathway that have altered TE in *clk-1(qm30)* versus *clk-1(qm30)*[*WT] represented in (C) and their involvement in longevity.

Besides OXPHOS transcripts, this study also identified an increased TE for the autophagy gene *lgg-1*, the *C. elegans* orthologue of LC3 in humans. Autophagy is an indispensable function in organisms undergoing lifespan extension. In the absence of *bec-1*, the *C. elegans* orthologue of the autophagy gene *APG6/VPS30/beclin1*, *daf-2*–mutant worms fail to undergo lifespan extension (Melendez et al, 2003). Reducing *bec-1* in worms subject to caloric restriction (*eat-2* mutants) or inhibition of the mTOR pathway

(reducing *let-363* [TOR] or *daf-15* [RAPTOR] expression) also has the same effect of preventing lifespan extension (Hansen et al, 2008). These findings are in line with our observation that mTOR is translationally repressed in *clk-1(qm30)* mutants (Fig 5D) and fits well with the notion that a reduction of growth can often lead to lifespan extension, in line with current theories proposing hyperfunction as a driving force in aging (Blagosklonny, 2006; Blagosklonny & Hall, 2009). Furthermore, our results suggest

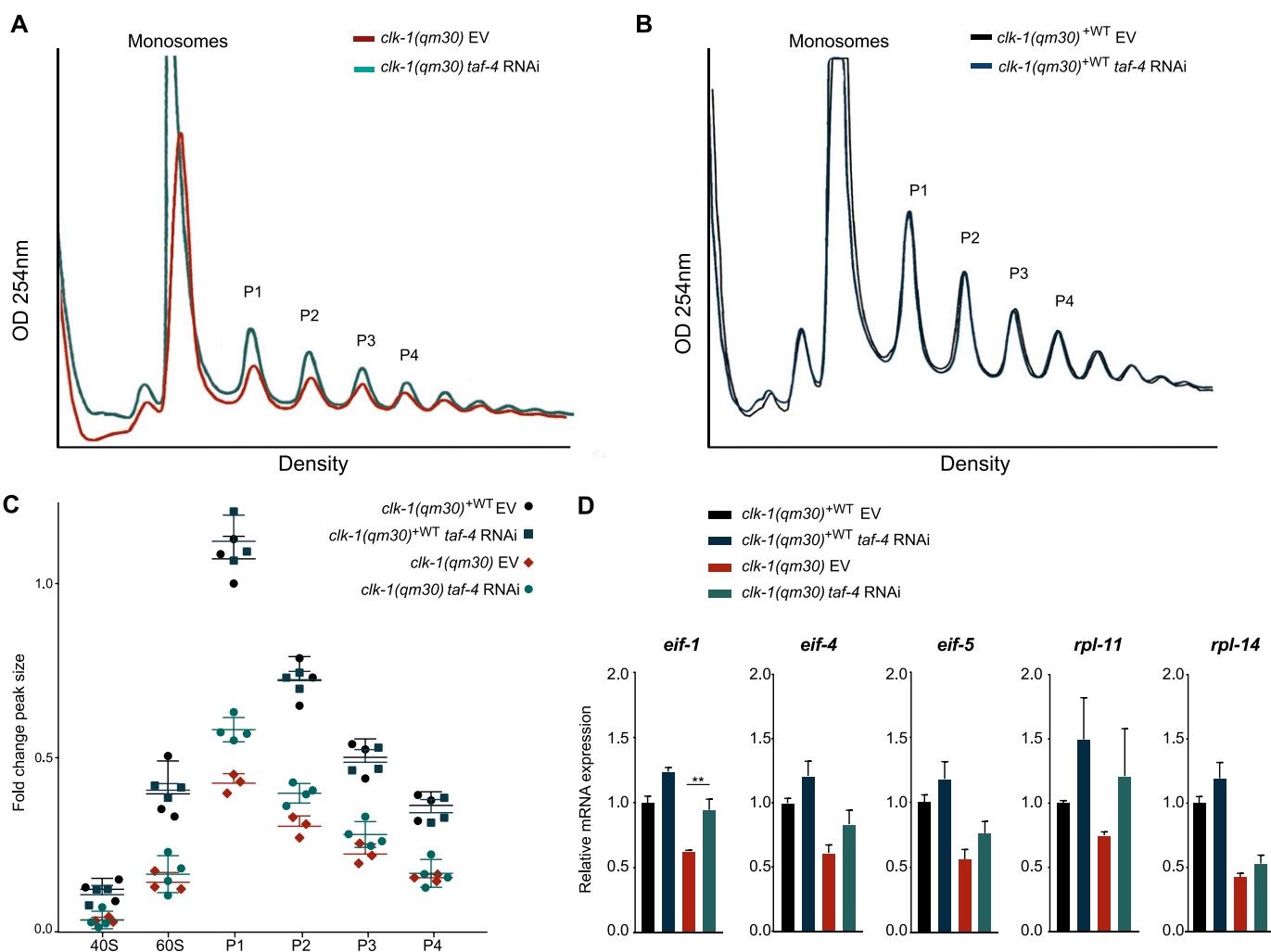

**Figure 6. RNAi of *taf*-4 partially restores repressed peaks in *clk-1(qm30)* mutants.**
**(A)** Representative traces of polysome profiles of *clk-1(qm30)* worms fed HT115 bacteria transformed with the empty vector (EV) or expressing *taf*-4 RNAi when harvested at the L4 stage, when lysate is normalized to total protein levels of 500 μg. The monosomal peak and polysomal peaks (P1–P5) are indicated. **(B)** Representative traces of polysome profiles of *clk-1(qm30)*[+WT] worms fed HT115 bacteria transformed with the EV or expressing *taf*-4 RNAi, when lysate is normalized to total protein levels of 500 μg. The monosomal peak and polysomal peaks are indicated. **(C)** Quantification of polysome peak sizes (AUC). The fold change is represented compared with P1 of the *clk-1(qm30)*[+WT] fed with HT115 bacteria. Polysomal peaks P1 and P2 were significantly different between *clk-1(qm30)*[+WT] fed with HT115 and *taf*-4 RNAi bacteria. No peaks were significantly different between *clk-1(qm30)*[+WT] fed with HT115 and *taf*-4 RNAi bacteria. Error bars represent mean ± SD. Significance was tested with t test and *P*-values were adjusted to correct for multiple testing using the Holm–Sidak method, with α = 0.05. **(D)** Relative expression levels determined by qRT–PCR in *clk-1(qm30)*[(+WT)] fed with either the EV or *taf*-4 RNAi bacteria. Expression levels were normalized using the geometrical mean of reference genes Y45F10D.4, *tba-1*, and *csq-1*. Significance was tested using one-way ANOVA with Sidak's multiple comparisons test. Error bars represent mean ± SEM. \*\**P* < 0.01, significance indicated only between *clk-1(qm30)* EV and *clk-1(qm30) taf*-4 RNAi.

a novel mechanism for inducing autophagy in systems undergoing lifespan extension by affecting the TE of mRNAs coding for crucial components of the mTOR-autophagy regulatory axis.

Finally, we found an unexpected role for TAF-4 as a repressor of polyribosome formation when UQ biosynthesis is compromised. Although TAF-4 is essential for all RNA transcriptions in the early embryo in *C. elegans* (Walker et al, 2001), this requirement is different in later developmental stages, and worms exposed to *taf*-4 RNAi starting from L1 throughout their remaining life grow normally (Khan et al, 2013). The authors propose a role for TAF-4 in the retrograde response model of longevity in mutants with dysfunctional mitochondria. In this model, dysfunctional mitochondria

signal via cAMP response element-binding proteins to activate TAF-4 in a way that increases life-extending gene expression. Here, we show that one of the consequences of TAF-4 loss in *clk-1(qm30)* mutants, and perhaps also in *isp-1* and *tpk-1* mutants, is a failure of the system that shuts down cytosolic mRNA translation in response to mitochondria impairment. This unexpected ability of TAF-4 to control protein synthesis, by regulating the expression of *eif-1*, will be very interesting for further study.

This study demonstrates that molecular pathways of long-lived *clk-1(qm30)* mutants with dysfunctional mitochondria converge with those of other distinctly different long-lived mutants to reduce cytoplasmic mRNA translation. It also reveals a striking change in the

transcripts that the polyribosomes prefer to translate when mitochondria are compromised. Ribosomal proteins and other protein parts of the translation machinery are very marginally translated, whereas the mitochondrial OXPHOS genes are still robustly translated despite a substantial reduction in the number of polysomes. We propose that this mechanism is likely recapitulated in many more models of lifespan extension and reaffirms studies that aim to extend human lifespan by reducing protein synthesis.

# Materials and Methods

## Nematode growing and conditions

*C. elegans* strains used MQ130 *clk-1(qm30)*, OL0100 ukSi1[p*clk-1*::*clk-1*::gfp, cb-unc-119(+)], and OL0120 ukSi2[p*clk-1*::*clk-1* ΔMTS (13-187)::gfp; cb-unc-119(+)] was kindly provided by AJ Whitmarsh and G Poulin (Monaghan et al, 2015) and *cguIs001* (FIB-1::GFP) was kindly provided by SJ Lo and T Ma (Lee et al, 2010). Worms were cultured at 20°C on nematode growth medium agar plates seeded with OP50 strain *Escherichia coli*. Synchronized worms were harvested and snap-frozen at the L4 larval stage for either total mRNA isolation or polysome profiling. For RNAi knockdown experiments, synchronized eggs were plated on nematode growth media containing IPTG (containing 2 mM IPTG) with a bacterial lawn of either *E. coli* HT115 (RNAi control strain, containing an empty vector) or *taf*-4 RNAi bacteria.

## Polysome profiling

Gradients of 17–50% sucrose (11 ml) in gradient buffer (110 mM KAc, 20 mM MgAc2, and 10 mM Hepes, pH 7.6) were prepared on the day before use in thin-walled, 13.2-ml, polyallomer 14 89-mm centrifuge tubes (Beckman Coulter). Nematodes were lysed in 500 ml polysome lysis buffer (gradient buffer containing 100 mM KCl, 10 mM MgCl2, 0.1% NP-40, 2 mM DTT, and RNaseOUT [Thermo Fisher Scientific]) using a Dounce homogenizer. The samples were centrifuged at 3,500 *g* for 10 min to remove debris and the supernatant was subjected to bicinchoninic acid assay (BCA) protein assay. In all, 500 mg of total protein for each sample was loaded on top of the sucrose gradients. The gradients were ultracentrifuged for 2 h at 40,000 rpm in a SW41Ti rotor (Beckman-Coulter). The gradients were displaced into a UA6 absorbance reader (Teledyne ISCO) using a syringe pump (Brandel) containing 60% sucrose. Absorbance was recorded at an optical density of 254 nm. All steps of this assay were performed at 4°C or on ice and all chemicals came from Sigma-Aldrich unless stated otherwise. Polysome peaks were quantified by measuring the area under the curve (AUC) in ImageJ. Significance was tested with *t* test, and *P*-values were adjusted to correct for multiple testing using the Holm–Sidak method, with $\alpha$ = 0.05.

## RNA sequencing

### Total/polysomal RNA isolation

For isolation of total RNA, worms were homogenized with a 5-mm steel bead using a TissueLyser II (Qiagen) for 5 min at a frequency of 30 times/s in the presence of TRIzol (Invitrogen), and the isolation was continued according to the manufacturer's protocol. Polysomal fractions from two experiments were pooled and mRNA was extracted using TRIzol LS (Invitrogen) according to the manufacturer's protocol. Contaminating genomic DNA was removed using RNase-Free DNase (Qiagen) and samples were cleaned up with the RNeasy MinElute Cleanup kit (Qiagen) and sequenced by GenomeScan BV at a 20 M read depth.

### Library preparation

The samples were processed for Illumina using the NEBNext Ultra Directional RNA Library Prep kit (NEB: New England Biolabs [NEB] #E7420) according to the manufacturer's protocol. Briefly, rRNA was depleted from total RNA using the rRNA depletion kit (NEB #E6310). After fragmentation of the rRNA-reduced RNA, a cDNA synthesis was performed to ligate with the sequencing adapters and PCR amplification of the resulting product. Quality and yield after sample preparation was measured with the Fragment Analyzer. Size of the resulting products was consistent with the expected size distribution (a broad peak between 300 and 500 bp). Clustering and DNA sequencing using the Illumina cBot and HiSeq 4000 was performed according to the manufacturer's protocol with a concentration of 3.0 nM of DNA. HiSeq control software HCS v3.4.0, image analysis, base calling, and quality check were performed with the Illumina data analysis pipeline RTA v2.7.7 and Bcl2fastq v2.17. RNA-seq data have been deposited in the ArrayExpress database at European Molecular Biology Laboratory: European Bioinformatics Institute under accession number: E-MTAB-7166.

## Bioinformatics analysis

### Mapping of the reads

The quality of the reads in the fastq files was confirmed using FastQC version 0.11.4 (http://www.bioinformatics.babraham.ac.uk/projects/fastqc). FastQC performs multiple quality tests and provides the user with warnings about possible problems with the raw data. Paired end reads were mapped to the *C. elegans* genome using HISAT2 version 2.1.0(Kim et al, 2015). The WBcel235 genome assembly was downloaded from WormBase and used as reference genome for the mapping. After successful mapping, counts per gene were extracted from the .SAM files with HTSeq version 0.9.1 (Anders et al, 2015). The -stranded=reverse setting was used, conforming with the NEB Ultra Directional RNA Library Prep kit procedure.

### Statistical analysis

The statistical analysis was performed in R using DESeq2 version 1.16.1 (Love et al, 2014). DESeq2 models the raw counts with the negative binomial distribution and normalizes between samples by calculating size factors using the median ratio method (Anders & Huber, 2010). The statistical test is based on a generalized linear model (GLM) using the negative binomial model as an error distribution model. To more accurately model dispersion parameters, the trend of dispersion to abundance is taken into account using an empirical Bayes procedure. The significance of the GLM coefficients is determined using a Wald test. The *P*-values are adjusted to

**Table 3. Primers for qPCR.**

| Gene | Gene ID | Forward primer | Reverse primer |
|---|---|---|---|
| Y45F10D.4 | 178344 | GTCGCTTCAAATCAGTTCAGC | GTTCTTGTCAAGTGATCCGACA |
| tba-1 | 172831 | AGACCAACAAGCCGATGGAG | TCCAGTGCGGATCTCATCAAC |
| csq-1 | 181563 | GTGACATCTAAATGGGCACGC | CTCACGGGTTTCCTCGTCAA |
| eif-1 | 266853 | TCAGCGTGACAAGGTCAAGG | GTGCACTCTGCAGTTGGACT |
| eif-3.C | 172858 | GGAGGACAAGGACAAGACGG | AGAAGGCTCGTGGCTTTTGA |
| inf-1 | 175966 | GGAAGGTCGACACACTCACC | ATGTCTCCGTGGAGGCAAGA |
| rpl-11.1 | 178778 | GGTTTCGGAGTTCAGGAGCA | TTGCGGTTCAGAACGACGTA |
| rpl-14 | 172818 | CAAGCTCACCGACTTCGAGA | GAGCTCCACTCGGACGATTC |

correct for multiple testing. In the differential translational regulation analysis, the fitted GLM took the following form:

$$\log(\mu_i) = \beta_0 + \beta_1 X_{sample} + \beta_2 X_{type} + \beta_3 X_{sample} \ X_{type}. \quad (1)$$

Here, the Wald test was performed on the interaction parameter 3 to infer significance. The log2 fold change calculated in this case corresponds to the following:

$$\log 2 \ \text{fold change} = \log 2 \left( \frac{\text{strain A}_{polysomal} / \text{strain A}_{total}}{\text{strain B}_{polysomal} / \text{strain B}_{total}} \right). \quad (2)$$

where in Equation (1), the sample refers to the strain and type to polysomal or total RNA. Experimental and statistical normalizations were performed for each condition individually, so counts for a specific gene in the polysomal dataset can be possibly higher than in the total RNA dataset, even though the first is technically a subset of the latter. Still, we can test if genes are highly translated in one strain when it is not in the other.

### Fluorescence microscopy imaging

For imaging, synchronized L4 stage N2 and *clk-1(qm30)* worms carrying FIB-1::GFP were placed on top of a 2% agarose pad on slides and anaesthetized using 1 mM levamisole. Nucleoli in the pharynx were imaged using 100× magnification (PLAN APO 100×/1.40/OIL Phaco3) with a Leica DM6 microscope and Leica LAS-X software. Total and average nucleolar area per worm was quantified using ImageJ.

### qPCR of total and polysomal RNA

cDNA synthesis was performed with 1 μg of total RNA using the QuantiTect Reverse Transcription Kit (Qiagen). LightCycler 480 SYBR Green I Master (Roche) was used for qPCR analysis, primers are listed in Table 3. Data were analyzed with LightCycler 480 software release 1.5 and LinRegPCR version 2015.3. Relative gene expression was normalized to the geometric mean of reference genes *Y45F10D.4, tba-1,* and *csq-1*.

## Supplementary Information

## Acknowledgements

The authors thank AJ Whitmarsh and G Poulin (University of Manchester) for providing the MQ130, OL0100, and OL0120 *C. elegans clk-1(qm30)* strains. We also thank SJ Lo and T Ma (Chang Gung University) for providing the FIB-1::GFP strain. Work in the Houtkooper group is financially supported by an European Research Council Starting grant (638290) and a VIDI grant from ZonMw (91715305) and the Velux Stiftung. Work in the Jelier group is financially supported by a Katholieke Universiteit Leuven grant C14/16/060. AW MacInnes and M Lezzerini are supported by E-Rare-2, the ERA-Net for Research on Rare Diseases (ZonMW 40-44000-98-1008). GE Janssens is supported by a 2017 Federation of European Biochemical Society long-term fellowship. We thank the financial support of the European Cooperation in Science and Technology initiative called "Group of C. elegans New Investigators in Europe (GENiE; BM1408)."

### Author Contributions

M Molenaars: conceptualization, formal analysis, investigation, visualization, methodology, and writing—original draft.
GE Janssens: data curation, formal analysis, funding acquisition, investigation, visualization, and writing—review and editing.
T Santermans: data curation, formal analysis, investigation, and writing—review and editing.
M Lezzerini: formal analysis, investigation, and writing—review and editing.
R Jelier: data curation, formal analysis, supervision, funding acquisition, investigation, and writing—review and editing.
AW MacInnes: conceptualization, formal analysis, supervision, funding acquisition, investigation, and writing—original draft.
RH Houtkooper: conceptualization, resources, formal analysis, supervision, funding acquisition, investigation, and writing—original draft.

### Conflict of Interest Statement

The authors declare that they have no conflict of interest.

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
