## [Reviewer comments · Life Science Alliance]

Mitochondrial ubiquinone-mediated longevity is marked by reduced cytoplasmic mRNA translation

Marte Molenaars, Georges E. Janssens, Toon Santermans, Marco Lezzerini, Rob Jelier, Alyson W. MacInnes, Riekelt H. Houtkooper
DOI: 10.26508/lsa.201800082

Review timeline:

Submission Date:	30 April 2018
Editorial Decision:	18 May 2018
Revision Received:	16 August 2018
Editorial Decision:	17 August 2018
Accepted:	23 August 2018

Report:

(Note: Letters and reports are not edited. The original formatting of letters and referee reports may not be reflected in this compilation.)

1st Editorial Decision

18 May 2018

Thank you for submitting your manuscript entitled "Mitochondrial ubiquinone-mediated longevity is marked by reduced cytoplasmic protein translation" to Life Science Alliance. The manuscript was assessed by expert reviewers, whose comments are appended to this letter. We invite you to submit a revision if you can address the reviewers' key concerns, as outlined here.

As you will see, reviewer #1 and #2 appreciate your data and support consideration of a revised version. Reviewer #3 acknowledges the technical quality of the work, but thinks that the study does not provide a sufficiently significant value to the field.

We have discussed your work within our editorial team and decided, given the support of two reviewers, that it is warranted to invite a revision.

We would appreciate if you could provide a point-by-point response to all criticisms raised and include text and figure changes accordingly, also addressing the request for providing information on statistical tests used. Only a few experiments that would further strengthen your work are suggested, and we would like to encourage you to follow the constructive input of the reviewers for the following:

- test translation efficient of specific genes (reviewer #1, point 1 (ii))
- measure nucleolar size of *clk-1* mutants using *fib-1::gfp* (reviewer #2)

Thank you for this interesting contribution to Life Science Alliance. We are looking forward to receiving your revised manuscript.

REFEREE REPORTS

Reviewer #1 (Comments to the Authors (Required)):

The study investigates the effect of mutation of the *clk-1* gene in *C. elegans* in promoting repression of protein translation. This is shown to occur via reduced polyribosome formation and a reduction in the translational efficiency of mRNAs encoding components of the translational machinery. This phenotype appears to be independent of the nuclear role of CLK-1 and is partially rescued by knock-down of the transcription factor TAF-4. As *clk-1* mutants have increased lifespan, the authors

propose that reduced protein translation may underpin this phenotype.

Overall, the study establishes that the loss of CLK-1 expression leads to a global decrease in protein translation. This is of interest as it indicates a potential signaling link between mitochondrial dysfunction and the regulation of protein translation leading to increased longevity.

Major points:

1. It remains unclear whether the altered translational response in *clk-1(qm30)* worms is a major contributor to their increased lifespan. The authors argue that TAF-4 is playing a key role in translational repression in *clk-1(qm30)* worms. However, the data supporting this is weak.
 - (i) The rescue effect of TAF-4 knock-down in the polyribosome profile in *clk-1(qm30)* worms is rather modest (Fig. 6C). Therefore, it is unclear whether this is making a significant contribution to rescuing the longevity phenotype or if the loss of other TAF-4 actions are responsible.
 - (ii) There is no evidence provided that TAF-4 knock-down specifically affects translation of mRNAs related to ribosome biogenesis and translation. This could be tested by determining if the translation efficiency of specific genes is altered.
2. A previously published microarray study compared mRNA expression patterns between *clk-1(qm30)* worms and N2 worms (Cristina et al., PLoS Genetics 5: e1000450). This study should be acknowledged and discussed within the context of the data presented in the current study.
3. In general, the figure legends are inadequate and lacking detail. These should be rewritten to provide much clearer descriptions of the data in the figures and the statistical tests employed.

Minor points:

- (i) Pg 5 - it is stated that, based on the RNA-seq analysis 'We next established the biological pathways involved in *clk-1* mediated longevity using DAVID and GO analysis'. This is overstated. The RNA-seq data simply identifies differences in transcript levels between the worms at L4 stage - it does not, in isolation, pinpoint those changes that are relevant to lifespan extension.
- (ii) Pg9, line 6 - reference is missing. Also, the Essers et al. reference in the reference list is lacking journal details.
- (iii) Throughout the manuscript, the *clk-1(qm30)* strain is often referred to as *clk-1(mq30)*. Also, the naming of worm strains is not consistent - for example the *clk-1(qm30)* strain with WT CLK-1 re-expressed is referred to as *clk-1(qm30)WT* in the text but as *clk-1 + WT* in some of the figures.
- (iv) Figures 5D and S2 do not appear to be mentioned in the text.

Reviewer #2 (Comments to the Authors (Required)):

General comments

The authors present a clear story on the interrelation between mitochondrial dysfunction, protein synthesis and longevity. In general, the study is well designed and conclusions are sound. It would have been more convincing to include alternative Mit mutations such as *isp-1* in the study to show the general trend of the reduction of mRNA translation of protein synthesis machinery components as a response to mitochondrial dysfunction. Although unlikely, this may be a specific phenotype restricted to *clk-1* mutants. However, I do not request such addition as this would take a complete redo of most experiments. One of the highlights in this study is the discovery of the potential new function of TAF-4 in translation control.

I would like to suggest a small straightforward orthogonal experiment that may further support the data presented here: measure nucleolar size of *clk-1* mutants using *fib-1::gfp*

(doi:10.1038/ncomms16083). *clk-1* nucleoli should be of smaller size than the control.

Figure 5A is not very intuitive and I needed a minute to understand this volcano plot, showing the log₂ values of a ratio of ratios (correctly defined by the authors as differential translational efficiency). However, it should be noted in the text that high differential translational efficiency should not, per definition, lead to high protein levels. One can imagine the example where *clk-1* worms, for a given gene, show a low level of total mRNAs compared to wild type (e.g. 25% of WT

levels). Even if translational efficiency would be 100%, absolute protein synthesis levels of that particular gene could be lower than wild type if the wild-type strain has a translational efficiency of 'only' 50%. In that sense, a proteomics approach could have given valuable additional information. Finally, I would suggest the authors to put their results in a broader context at the very end of the discussion (e.g. their data fits well with the notion that a reduction of growth leads to lifespan extension, supporting the hyperfunction theory of aging).

Specific comments

Title: this is nitpicking but 'protein translation' is a contamination of 'protein synthesis' and 'mRNA translation' (the protein does not get translated). I realize this term is often used, but it is not correct. Use italics for *clk-1* consistently throughout the manuscript. Also use the WT superscript consistently.

Page 7 last line: mitochondria (not mitochondrial)

Page 9, paragraph 2, line 2: insert appropriate reference (Kahn et al., 2013)

Page 9, paragraph 2, line 7: the polysome (not te polysome)

Page 9, paragraph 2, line 8: ...that TAF-4 loss partially restores polyribosome formation...

Figure 4B, horizontal axis: *clk-1* total RNA, fold change vs WT (log₂)

Reviewer #3 (Comments to the Authors (Required)):

In the manuscript entitled "Mitochondrial ubiquinone-mediated longevity is marked by reduced cytoplasmic protein translation", Houtkooper and collaborators show that mutations in the *clk-1* gene (known to negatively affect ubiquinone synthesis and extend lifespan) also lead to a reduced translational rate in the cytoplasm. This reduction is partially relieved when the TAF-4 transcription factor is knocked down by RNAi. The analysis involved polysome analysis, RNA-seq and RT-qPCR experiments on wild type animals, *clk-1* mutants and *clk-1* mutant rescued by a nuclear version of *clk-1*. From a technical perspective, this body of work seems to be well executed.

Results show that *clk-1* mutant harbor an overall reduced number of transcripts and more specially involved in certain processes such as oxido-reduction, reproduction and development and protein translation and mRNA processing. No obvious difference was observed between a *clk-1* mutant with no rescue and animals where CLK-1 was specifically re-introduced in the nucleus. Polysome profiling on these animals also shows a repressed protein translation and this repression is somewhat relieved when TAF-4 is knocked down by RNAi.

Although the techniques used and the data produced seems robust, this works provides little novelty, given that the authors have already published that polysome analysis of *clk-1* (Essers et al, Cell Reports). Indeed, we already knew that *clk-1* exhibited an altered polysomal profile and data presented here basically comfort this view. The involvement of TAF-4 can be considered as novel, but although interesting, this finding does not alter greatly the way we think about lifespan regulation or even the way CLK-1 works.

It is interesting that the nuclear version of CLK-1 makes little difference, bringing argument against a strong impact of *clk-1* in the nucleus. This debate was going on.

My conclusion is that although the work submitted by Molenaars et al is technically sound, the novelty that it brings is very limited. For this reason, I would recommend to produce more important scientific insights before considering publishing this body of work. Only when the work brings something really different from Essers et al., will it be interesting to publish. An alternative way to leading with these data would be to focus on the absence of evidence that nuclear CLK-1 play a significant role.

In addition, there are a series of details such as allele names that are not written in italics as it should and a missing reference on page 9 that should be corrected.

Reviewer #1:

The study investigates the effect of mutation of the *clk-1* gene in *C. elegans* in promoting repression of protein translation. This is shown to occur via reduced polyribosome formation and a reduction in the translational efficiency of mRNAs encoding components of the translational machinery. This phenotype appears to be independent of the nuclear role of CLK-1 and is partially rescued by knock-

down of the transcription factor TAF-4. As *clk-1* mutants have increased lifespan, the authors propose that reduced protein translation may underpin this phenotype. Overall, the study establishes that the loss of CLK-1 expression leads to a global decrease in protein translation. This is of interest as it indicates a potential signaling link between mitochondrial dysfunction and the regulation of protein translation leading to increased longevity.

We thank the reviewer for the constructive evaluation of our manuscript.
Major points:

1. It remains unclear whether the altered translational response in *clk-1(qm30)* worms is a major contributor to their increased lifespan.

Longer-lived *C. elegans* mutants are being increasingly shown to be marked by reduced mRNA translation (doi: 10.1038/msb.2013.35, 10.1186/1755-8794-1-33). The fact that reduction in mRNA translation itself (by knockdown of factors involved in the translation machinery) leads to lifespan extension shows that it indeed plays a key role in longevity.

Another way we now make this connection in the revision is through a suggestion of Reviewer 2, using the established hallmark of longevity that is nucleolar size (doi:10.1038/ncomms16083). The nucleoli facilitate ribosome biogenesis, the rate of which has a direct impact on translation. In longer-lived mutants the nucleoli appear smaller (doi:10.1038/ncomms16083), suggesting a reduction in ribosome biogenesis that in turn reduces translation. In order to make the connection between the *clk-1(qm30)* worms, ribosome biogenesis, protein translation, and longevity more convincing, we measured nucleolar size in these worms by using the FIB-1::GFP reporter. Here we show that the nucleoli of the *clk-1(qm30)* worms are indeed smaller compared to N2 worms (Fig 4F-G), in line with other long-lived *C. elegans* models (doi:10.1038/ncomms16083). Furthermore, when looking at *fib-1* expression in our RNA-seq data, we observed a marked downregulation of *fib-1* in both the total and the polysomal RNA of the *clk-1(qm30)* compared to the *clk-1(qm30)+WT* worms (Fig 4E).

The authors argue that TAF-4 is playing a key role in translational repression in *clk-1(qm30)* worms. However, the data supporting this is weak. (i) The rescue effect of TAF-4 knock-down in the polyribosome profile in *clk-1(qm30)* worms is rather modest (Fig. 6C). Therefore, it is unclear whether this is making a significant contribution to rescuing the longevity phenotype or if the loss of other TAF-4 actions are responsible.

RNAi of *taf-4* did not fully reverse extended lifespan of *clk-1(qm30)* (doi:10.18632/aging.100604). We think our results showing partial rescue of polysome profiles following *taf-4* RNAi results are in line with this. What we would also like to point out is that RNAi does not knock out *taf-4* but knocks it down (by 60%).

(ii) There is no evidence provided that TAF-4 knock-down specifically affects translation of mRNAs related to ribosome biogenesis and translation. This could be tested by determining if the translation efficiency of specific genes is altered.

Since TAF-4 is a transcription factor, we would expect upregulation in the total RNA rather than changing the translational efficiency of specific genes. We agree that looking at mRNAs related to translation in the TAF-4 knockdown condition is very interesting, as suggested. To address this, we have measured the expression of a selected set of genes encoding proteins involved in translation upon *taf-4* RNAi in the *clk-1(qm30)* mutants by qPCR. We show that the translation initiation factor *eif-1* is significantly upregulated upon *taf-4* RNAi in the *clk-1(qm30)* mutant compared to the empty vector control, and even restored the expression back to *clk-1(qm30)+WT* levels. *Eif-1* is important for translation initiation and yeast *eif-1* mutants (*sui1-*) show a reduction of polysomes compared to wildtype (doi: 10.1128/MCB.12.1.248). Moreover, the reduction of *eif-1* extends lifespan in N2 worms (doi:10.1371/journal.pgen.1001048). Other genes, such as *eif-3c*, *eif-4*, *eif-5*, *rpl-11* and *rpl-14* show a similar trend but without reaching significance. These data are included as Fig 6D of our revised manuscript.

2. A previously published microarray study compared mRNA expression patterns between *clk-1(qm30)* worms and N2 worms (Cristina et al., PLoS Genetics 5: e1000450). This study should be acknowledged and discussed within the context of the data presented in the current study.

We have included this paper in our discussion (page 10).³ In general, the figure legends are inadequate and lacking detail. These should be rewritten to provide much clearer descriptions of the data in the figures and the statistical tests employed.

We added more detail to the figure legends, including description of the statistical tests employed.

Minor points:

(i) Pg 5 - it is stated that, based on the RNA-seq analysis 'We next established the biological pathways involved in clk-1 mediated longevity using DAVID and GO analysis'. This is overstated. The RNA-seq data simply identifies differences in transcript levels between the worms at L4 stage - it does not, in isolation, pinpoint those changes that are relevant to lifespan extension. We rephrased this sentence in order to make it more accurate as suggested.

(ii) Pg9, line 6 - reference is missing. Also, the Essers et al. reference in the reference list is lacking journal details. This has been corrected.

(iii) Throughout the manuscript, the clk-1(qm30) strain is often referred to as clk-1(mq30). Also, the naming of worm strains is not consistent - for example the clk-1(qm30) strain with WT CLK-1 re-expressed is referred to as clk-1(qm30)WT in the text but as clk-1 + WT in some of the figures. We carefully checked and corrected these mistakes.

(iv) Figures 5D and S2 do not appear to be mentioned in the text. Figure 5D and S2 are now referred to in the text.

Reviewer #2: General comments

The authors present a clear story on the interrelation between mitochondrial dysfunction, protein synthesis and longevity. In general, the study is well designed and conclusions are sound. It would have been more convincing to include alternative Mit mutations such as *isp-1* in the study to show the general trend of the reduction of mRNA translation of protein synthesis machinery components as a response to mitochondrial dysfunction. Although unlikely, this may be a specific phenotype restricted to *clk-1* mutants. However, I do not request such addition as this would take a complete redo of most experiments. One of the highlights in this study is the discovery of the potential new function of TAF-4 in translation control.

We thank the reviewer very much for the positive comments about the study, its design, and the conclusions.

I would like to suggest a small straightforward orthogonal experiment that may further support the data presented here: measure nucleolar size of *clk-1* mutants using *fib-1::gfp* (doi:10.1038/ncomms16083). *clk-1* nucleoli should be of smaller size than the control.

This is an interesting point and we agree with the reviewer that smaller nucleoli are expected in the *clk-1*(qm30) mutants. Ribosome biogenesis takes place in the nucleoli and our data already showed that ribosome biogenesis is downregulated in the *clk-1*(qm30) worms (Fig 4D). We looked further in our RNAseq and noticed clear downregulation of *fib-1* in both the total RNA and the polysomal RNA of *clk-1* (Fig 4E). As suggested, we also measured nucleolar size in the *clk-1* mutants using the FIB-1::GFP nucleoli reporter. In our revised manuscript, we show that the nucleoli of the *clk-1*(qm30) worms are indeed smaller compared to N2 worms (new fig 4F,G) similar to what has been observed in other long-lived *C. elegans* models (doi:10.1038/ncomms16083).

Figure 5A is not very intuitive and I needed a minute to understand this volcano plot, showing the log₂ values of a ratio of ratios (correctly defined by the authors as differential translational efficiency).

We understand that a ratio of ratios (differential translational efficiency), may not be very intuitive at first glance. Therefore, we have added explanatory panels to the figure, corresponding to an enrichment of transcripts in the polysomal fraction (high TE) or a depletion of it (low TE). With

these we aim to have the concept of TE be more accessible to readers and thank the reviewer for this point.

However, it should be noted in the text that high differential translational efficiency should not, per definition, lead to high protein levels. One can imagine the example where *clk-1* worms, for a given gene, show a low level of total mRNAs compared to wild type (e.g. 25% of WT levels). Even if translational efficiency would be 100%, absolute protein synthesis levels of that particular gene could be lower than wild type if the wild-type strain has a translational efficiency of 'only' 50%. In that sense, a proteomics approach could have given valuable additional information.

We agree with the reviewer that TE is not the same as absolute protein synthesis and we do not wish to give this impression in our manuscript. The changes in TE are still biologically relevant since there is preferred translation of certain mRNA despite of the mRNA availability and overall reduced levels of translation in the context of *clk-1*(qm30) worms.

Finally, I would suggest the authors to put their results in a broader context at the very end of the discussion (e.g. their data fits well with the notion that a reduction of growth leads to lifespan extension, supporting the hyperfunction theory of aging).

Thank you for this suggestion. To emphasize this more we have added a piece in the discussion on page 12-13, referencing both our observations of mTOR suppression (Fig 5D), and the hyperfunction theory of aging.

Specific comments

Title: this is nitpicking but 'protein translation' is a contamination of 'protein synthesis' and 'mRNA translation' (the protein does not get translated). I realize this term is often used, but it is not correct. We have replaced the term "protein translation" with "protein synthesis" or "mRNA translation" where appropriate.

Use italics for *clk-1* consistently throughout the manuscript. Also use the WT superscript consistently.

We carefully checked and corrected these mistakes.

Page 7 last line: mitochondria (not mitochondrial)
This has been corrected.

Page 9, paragraph 2, line 2: insert appropriate reference (Kahn et al., 2013)
This has been corrected.

Page 9, paragraph 2, line 7: the polysome (not te polysome)
This has been corrected.

Page 9, paragraph 2, line 8: ...that TAF-4 loss partially restores polyribosome formation...
This has been corrected.

Figure 4B, horizontal axis: *clk-1* total RNA, fold change vs WT (log₂)
This has been corrected.

Reviewer #3:

In the manuscript entitled "Mitochondrial ubiquinone-mediated longevity is marked by reduced cytoplasmic protein translation", Houtkooper and collaborators show that mutations in the *clk-1* gene (known to negatively affect ubiquinone synthesis and extend lifespan) also lead to a reduced translational rate in the cytoplasm. This reduction is partially relieved when the TAF-4 transcription factor is knocked down by RNAi. The analysis involved polysome analysis, RNA-seq and RT-qPCR experiments on wild type animals, *clk-1* mutants and *clk-1* mutant rescued by a nuclear version of *clk-1*. From a technical perspective, this body of work seems to be well executed. We thank the reviewer for his/her comments, and we appreciate the positive evaluation of the technical aspect of our work.

Results show that *clk-1* mutant harbor an overall reduced number of transcripts and more specially involved in certain processes such as oxido-reduction, reproduction and development and protein translation and mRNA processing. No obvious difference was observed between a *clk-1* mutant with no rescue and animals where CLK-1 was specifically re-introduced in the nucleus. Polysome profiling on these animals also shows a repressed protein translation and this repression is somewhat relieved when TAF-4 is knocked down by RNAi.

Although the techniques used and the data produced seems robust, this works provides little novelty, given that the authors have already published that polysome analysis of *clk-1* (Essers et al, Cell Reports). Indeed, we already knew that *clk-1* exhibited an altered polysomal profile and data presented here basically comfort this view. The involvement of TAF-4 can be considered as novel, but although interesting, this finding does not alter greatly the way we think about lifespan regulation or even the way CLK-1 works. It is interesting that the nuclear version of CLK-1 makes little difference, bringing argument against a strong impact of *clk-1* in the nucleus. This debate was going on.

We respectfully disagree that our paper lacks novelty. It is true that the polysome analysis of *clk-1* (*qm30*) is reported as a supplemental figure in Essers et al, Cell Reports (doi: 10.1016/j.celrep.2014.12.029). However, the polysome analysis on its own does not give any information on 1) the cause of the change in polysomal profiles, and 2) what messages are preferably translated during this altered translational response. These remaining scientific insights, which are far more informative than only polysome profiles, were both answered in this manuscript and cover the main part of our manuscript.

Moreover, the Essers paper focuses its in-depth analysis exclusively on the *daf-2* mutant that has impaired insulin signaling. This is a fundamentally different pathway than the one we are studying with the *clk-1* (*qm30*) mutants that have impaired ubiquinone biosynthesis. So, although the final impact on protein synthesis may be similar, our work is still novel in showing that inhibition of mitochondrial function results in such a dramatic reduction of the translation machinery. To the best of our knowledge this connection has not been previously made.

My conclusion is that although the work submitted by Molenaars et al is technically sound, the novelty that it brings is very limited. For this reason, I would recommend to produce more important scientific insights before considering publishing this body of work. Only when the work brings something really different from Essers et al., will it be interesting to publish. An alternative way to leading with these data would be to focus on the absence of evidence that nuclear CLK-1 play a significant role.

Please see our arguments above that we do in fact provide novel insights into how mitochondrial dysfunction (not reduced insulin signaling) is driving lifespan extension through a reduction of the translation machinery. Such cross-organellar communication to manage specific mRNA translation is highly novel. We would again like to point out that our interest focused on the cause of the changed translation rates and the effect on the TE of individual mRNAs. Although we do not feel that this study should be the final arbiter of the role of the alleged nuclear form of CLK-1, we do make a point of discussing this lack of evidence in depth on page 11. This aside, we do feel that we have established an interesting and novel link between mitochondrial dysfunction, a reduction of protein synthesis, and longevity in eukaryotes.

In addition, there are a series of details such as allele names that are not written in italics as it should and a missing reference on page 9 that should be corrected.

We have corrected them throughout the text. Also, the missing reference is corrected.

2nd Editorial Decision

17 August 2018

Thank you for submitting your revised manuscript entitled "Mitochondrial ubiquinone-mediated longevity is marked by reduced cytoplasmic mRNA translation". We appreciate the introduced changes and would be happy to publish your paper in Life Science Alliance pending final minor revisions:

- we think it would be good to provide for figure 4F/G a zoomed image to allow others to more easily appreciate the difference in nucleolar size observed
- please add a callout to figure S1
